# Drebrin-mediated microtubule–actomyosin coupling steers cerebellar granule neuron nucleokinesis and migration pathway selection

Niraj Trivedi[1,*], Daniel R. Stabley[1,*], Blake Cain[1,*], Danielle Howell[1,*], Christophe Laumonnerie[1,*], Joseph S. Ramahi[1], Jamshid Temirov[2], Ryan A. Kerekes[3], Phillip R. Gordon-Weeks[4] & David J. Solecki[1]

Neuronal migration from a germinal zone to a final laminar position is essential for the morphogenesis of neuronal circuits. While it is hypothesized that microtubule–actomyosin crosstalk is required for a neuron's 'two-stroke' nucleokinesis cycle, the molecular mechanisms controlling such crosstalk are not defined. By using the drebrin microtubule–actin crosslinking protein as an entry point into the cerebellar granule neuron system in combination with super-resolution microscopy, we investigate how these cytoskeletal systems interface during migration. Lattice light-sheet and structured illumination microscopy reveal a proximal leading process nanoscale architecture wherein f-actin and drebrin intervene between microtubules and the plasma membrane. Functional perturbations of drebrin demonstrate that proximal leading process microtubule–actomyosin coupling steers the direction of centrosome and somal migration, as well as the switch from tangential to radial migration. Finally, the Siah2 E3 ubiquitin ligase antagonizes drebrin function, suggesting a model for control of the microtubule–actomyosin interfaces during neuronal differentiation.

[1] Department of Developmental Neurobiology, St Jude Children's Research Hospital, 262 Danny Thomas Place, Memphis, Tennessee 38105, USA. [2] Cell & Tissue Imaging Center, St Jude Children's Research Hospital, 262 Danny Thomas Place, Memphis, Tennessee 38105, USA. [3] Imaging, Signals and Machine Learning Group, Oak Ridge National Laboratory, Oak Ridge, Tennessee 37831, USA. [4] Medical Research Council MRC Centre for Developmental Neurobiology, King's College London, London SE1 1UL, UK. * These authors contributed equally to this work. Correspondence and requests for materials should be addressed to D.J.S. (email: david.solecki@stjude.org).

Understanding how neurons migrate from germinal zone (GZ) niches to their final laminar positions while synchronizing morphology to cytoskeletal organization is a classic problem in developmental neurobiology. The clinical significance of neuronal migration is illustrated by the spectrum of neurological disorders in which lamination defects arising from disrupted neuronal motility lead to intellectual disabilities and epilepsy[1–4]. Two recent conceptual advances have coalesced into an overall framework that guides cellular and molecular investigations of migration[5–9]. First, most mammalian neurons undergo a stereotyped saltatory movement cycle as they migrate to their final laminae, despite the apparent diversity in migration substrates, adhesive receptor systems, guidance cues and migration pathways[10–13]. Each neuronal movement cycle begins with a leading process extending in the direction of migration[14,15], followed by the transport of cytoplasmic organelles, such as the centrosome or Golgi apparatus, into an engorged proximal segment of the leading process, and ends with the directed nuclear movement termed nucleokinesis[16–23]. Second, insights gained from human neuronal migration disorders and genetic manipulations in mice highlight the importance of the neuronal cytoskeleton or cytoskeletal adaptors in producing the stereotyped motility patterns necessary for appropriate neuronal migration during development[2,3,24–26].

Time-lapse imaging of elements of the microtubule or actin cytoskeletons has provided a detailed spatiotemporal framework for the two-stroke motility cycle. During the first phase of the cycle, microtubule growth and bulk movement of centrosomal and noncentrosomal microtubules is oriented towards the proximal leading process[17,19,20,27–30], where cytoplasmic organelles, including the centrosome, translocate. Although cytoplasmic dynein within the proximal leading process has been implicated in microtubule and centrosome movement in the direction of migration[19,31], it remains unclear where microtubules are anchored and how pulling forces are transmitted to the microtubule cytoskeleton. Surprisingly, after an initial focus on the role of actomyosin in the rear of migrating neurons[18,19,32–34], time-lapse imaging revealed that the leading process is a major site for f-actin dynamics and myosin II motor activity[23,35–39]. Traction force microscopy shows that discrete leading process actomyosin contraction centres exert force before somal translocation, generating traction force between a neuron and its migration substrate[40]. Moreover, myosin II-powered actin flow from the proximal to distal leading process drives centrosome positioning, adhesion and/or guidance receptor dynamics[23,35–37]. Thus, actomyosin simultaneously coordinates external traction with polarized organelle transport in the migrating neuron.

Despite the definition of the individual contributions of the microtubule and actin cytoskeletons to two-stroke motility, our knowledge of how the two systems function together in migrating neurons remains fragmentary. Here we examined the cell biological underpinnings of microtubule–actomyosin crosstalk in cerebellar granule neurons (CGNs) by utilizing drebrin, a protein that links microtubule movements to the actomyosin cytoskeleton in the growth cone and synapse[41–43], as a molecular entry point[44]. Drebrin is required for neuronal migration in the mouse olfactory systems and chick spinal cord[45,46]. We applied advanced lattice light-sheet (LLS) microscopy and Super-resolution structured illumination microscopy (SR-SIM) in combination with functional studies examining the dynamic localization and function of drebrin in migrating CGNs. These studies reveal that drebrin and f-actin in the proximal leading process/cytoplasmic dilation constitute a novel interaction interface directing movements of the

centrosome and microtubules that steer the polarity of the two-stroke motility cycle. We also define a novel functional antagonism between drebrin and the Siah2 E3 ubiquitin ligase that illustrates how the cytoskeletal underpinnings of the two-stroke motility cycle are regulated during neuronal differentiation.

## Results

**Drebrin reports leading process microtubule-actin interfaces.** In CGNs, both the actin and microtubule cytoskeletal motor systems contribute to two-stroke motility (Supplementary Note 1, Supplementary Figs 1 and 2); thus, we explored functional microtubule–actomyosin interactions as a means to define mechanistically how both systems work together to impact neuronal motility. We reasoned that drebrin would be a useful reporter for establishing the sites of microtubule–actomyosin interaction in migrating CGNs, as it binds the sides of f-actin filaments and microtubule plus ends through direct interaction with the neuronal +TIP protein end-binding protein 3 (EB3)[41,47]. Drebrin mediates f-actin–microtubule interactions in growth cones or dendritic spines and is required for some forms of neuronal migration, but its dynamic localization had not been examined in migrating neurons, nor had its function been examined in the two-stroke motility model. We examined drebrin expression in the developing cerebellum to determine its suitability as a reporter. Consistent with previous drebrin expression analyses[48–50], at postnatal day 7 (P7) drebrin displays a complementary expression profile with the granule neuron progenitor (GNP) markers Ki67 and Zeb1 and is expressed in the cerebellar lamina where differentiated CGNs reside (for example, the inner external granular layer (EGL), molecular layer and inner granular layer (IGL), (Fig. 1a,b). CGN drebrin expression is linked to terminal differentiation, as it commences in CGNs that co-express the cyclin-dependent kinase inhibitory protein p27Kip1/Cdkn1b (p27 hereafter), a marker for newly postmitotic CGNs in the inner EGL (Fig. 1c). The drebrin E isoform is the predominant drebrin protein expressed at this developmental stage as little drebrin A immunoreactivity is present in CGNs, consistent with previous expression analyses (Fig. 1d). Immunohistochemical staining of purified CGNs from P7 cerebellum revealed endogenous drebrin protein in the neuronal soma and leading process, where it was largely coincident with myosin II motors and microtubules in important regions of migrating CGNs, such as the proximal leading process (Fig. 1e,f). The leading process and soma also showed immunoreactivity for drebrin phosphorylated on serine 142, which indicates an active configuration to interact with the actin and microtubule cytoskeletons (Fig. 1g and Supplementary Fig. 3 for detailed phospho-Ser142 analysis)[47].

Previous studies have revealed a proximal-to-distal f-actin flow from the neuronal soma through the proximal leading process/cytoplasmic dilation that accompanies neuronal migration[23,35–39,51]. This flow is required for centrosomal positioning in the first phase of the two-stroke motility cycle[35]. We hypothesized that if f-actin–microtubule interactions occur in the proximal leading process, then f-actin–microtubule crosslinking factors might be recruited there during the motility cycle. Accordingly, we generated a fluorescent drebrin E reporter for use in live-cell imaging experiments by fusing a tandem repeat of Kusabira Orange (KO1) to the 3′ end of the mouse full-length drebrin E complementary DNA. We monitored drebrin dynamics in a well-established in vitro CGN–Bergmann glial model system for radial migration by imaging CGNs expressing drebrin E-2xKO1 and Map2c-yellow fluorescent protein (Map2c-YFP) with a spinning disk confocal microscope. A custom-written

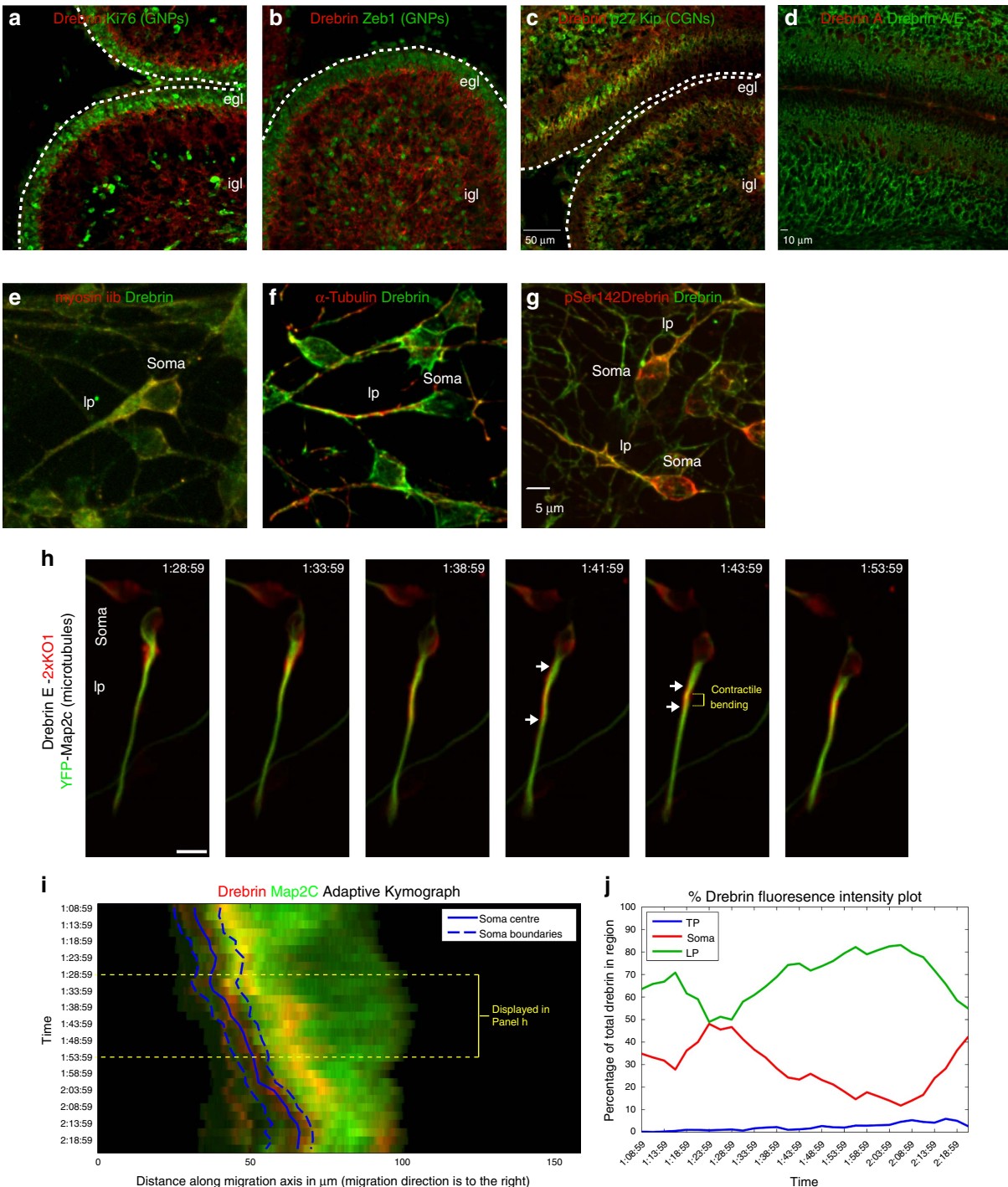

**Figure 1 | Drebrin protein is expressed in differentiated CGNs and is dynamically localized to the leading process during two-stroke motility.**
(**a**–**c**) Immunohistochemical examination of drebrin expression in the P7 mouse cerebellum. In the P7 cerebellum, drebrin (red) is complementarily expressed with the GNP markers Ki67 (green) (**a**) or Zeb1 (green) (**b**) and is co-expressed in differentiated CGNs with p27Kip (green) (**c**), scale bar for each image equals 50 µm. Drebrin A expression (red) is minimal at P7 (**d**), scale bar, 10 µm. (**e**–**g**) Immunocytochemical examination of drebrin expression in CGNs grown in culture. Expression of drebrin protein (green) is coincident with myosin IIB, alpha-tubulin, and drebrin phospho-Ser142 immunoreactivity (all red) (lp = leading process), scale bar for each equals 5 µm. (**h**) Drebrin (labelled with drebrin E 2x-KO1, red) is initially localized in the cell body but becomes restricted to the leading process f-actin domain, where it appears to contract around the time of somal translocation. Around the time of local contraction, bending is observed in the microtubule cytoskeleton (labelled by YFP-Map2C, white arrows). Scale bar, 10 µm. Time stamp = hours:min:sec. (**i**) Adaptive volumetric kymographs of the sequences shown in **h**: the leading process and direction of migration is towards the right of the panel (dashed blue line = somal boundaries; solid blue line = soma centre; red = drebrin E 2x-KO1; green = YFP-Map2C). The yellow box highlights the time-lapse frames shown in **h**, in which a subpopulation of drebrin flows down the leading process in the direction of migration before the contraction of the leading process drebrin domain. Arrows indicate the borders of the drebrin domain as it contracts during a migration cycle. (**j**) Analysis of the percentage of regional drebrin localization signal in the soma and leading or trailing process in migrating neurons expressing drebrin E 2x-KO1, in which drebrin accumulates in the leading process in the active phase of migration.

adaptive kymograph algorithm[23] showed that drebrin E-2xKO1 is cyclically transferred from the neuronal soma to the proximal portion of the leading process during CGN migration with no significant fluctuations in the trailing process, much as we previously observed for f-actin[35] (Fig. 1h–j). The drebrin E-labelled domain contracts during migration (compare panel 1:41:59 with 1:43:59 in Fig. 1h), and bending can be seen in the Map2c-YFP-labelled microtubule bundles in the centre of the contractile region. Interestingly, treatment with the blebbistatin myosin II inhibitor halts drebrin translocation, suggesting that leading process actomyosin flow contributes to drebrin translocation (Supplementary Fig. 4). These results show that drebrin is expressed in CGNs after they differentiate, with timing similar to that of the radial migration initiation. Moreover, drebrin dynamically localizes to the proximal leading process before or during somal translocation and is the clearest reporter yet for detecting actomyosin component transference from the soma to the leading process during the two-stroke migratory cycle.

**LLS and SR-SIM reveal leading process nanoscale structures.** The images of endogenous drebrin and microtubules in our immunocytochemically stained fixed CGNs (Fig. 1f) and of drebrin E-2xKO1 and Map2c-YFP in live CGNs indicated that, in the proximal leading process, drebrin was located in a layer outside the more central microtubule structures. However, the resolution of standard confocal microscopy is insufficient to support such a conclusion. Therefore, we applied LLS microscopy, which utilizes ultrathin light sheets derived from optical lattices to rapidly section living cells with minimal phototoxicity and provide near isotropic imaging in the $x$, $y$ and $z$ dimensions[52]. For example, the XZ sectioning of LLS is sufficient to resolve the plasma membrane interface of a CGN interacting with a glial cell in culture, which had previously only been examined in fixed cells using electron microscopy[27] (Supplementary Videos 1 and 2). Time-lapse imaging of CGNs expressing drebrin E-2xKO1 and glycophosphatidylinositol-pHluorin (GPI-pHluorin), a novel probe that exploits the pH sensitivity of super-ecliptic pHluorin to label only the exterior surface of the plasma membrane[53], confirmed that drebrin flows down the CGN leading process before somal translocation but does not fully extend to its distal tip (Fig. 2a, Supplementary Video 3). With its superior signal-to-noise ratio and spatial resolution (enhanced by Lucy–Richardson deconvolution using measured point-spread functions), LLS microscopy revealed that drebrin is localized to two sub-plasma membrane collars parallel to the long axis of the CGN leading process that were difficult to observe by spinning disk confocal microscopy. We should note that not all CGNs are synchronized to the same stage of the migration cycle and can stochastically change migration directions along glial fibres *in vitro* leading to minor variability in the appearance of drebrin in the trailing process. We next used an enhanced green fluorescent protein-labelled utrophin actin-binding domain (EGFP-UTRCH-ABD) as a reporter of f-actin localization[54]. During CGN migration, levels of EGFP-UTRCH-ABD in the neuronal soma and leading process were high, and it localized to a cortical collar (Fig. 2b, Supplementary Video 4). Although the leading process f-actin signal was evident before somal translocation (see time-point 8:47 in Fig. 2b) and became enriched in the proximal leading process, drebrin E-2xKO1 became most clearly coincident with f-actin when it enriched in the proximal leading process during movement (see time-point 22:40 in Fig. 2b). Imaging drebrin E-2xKO1 in combination with Tau-Emerald, a microtubule marker, revealed that as drebrin became enriched in the dilated proximal leading process, it

appeared to envelop a central core of microtubules (Fig. 2c, Supplementary Video 5). Finally, time-lapse imaging of CGNs expressing pHluorin-tagged JAM-C, an immunoglobulin superfamily adhesion molecule essential for CGN migration[55], and drebrin E-2xKO1 revealed surface JAM-C at punctate junctions within the leading process and neuronal soma (Fig. 2d, Supplementary Video 6). Although many adhesive puncta were present in the distal leading process, a subset was located in the drebrin E-enriched portion of the proximal leading process, suggesting that these adhesions are substrate contact points for contractile forces generated in the drebrin E-enriched domain.

SR-SIM improves $x$, $y$ and $z$ resolution two-fold compared with standard forms of diffraction-limited microscopy[56–58] and is thus ideal to extend our analysis of the proximal leading process. To examine the relationship of drebrin in the proximal leading process to the plasma membrane, f-actin and microtubules, we developed a live-cell labelling protocol that involved imaging two genetically encoded fluorescent probes (drebrin E-2xKO1 and EGFP-UTRCH-ABD or GPI-pHluorin) in combination with the far-red SiR-tubulin microtubule dye. This protocol avoided artefacts associated with cytoskeletal fixation and used a commercial SR-SIM system with a resolution of ∼110 nm. Imaging CGNs with dilated proximal leading processes treated with two label combinations (GPI-pHluorin/drebrin E-2xKO1/SiR-tubulin or EGFP-UTRCH-ABD/drebrin E-2xKO1/SiR-tubulin) revealed that (1) the proximal leading process does indeed possess a central core of microtubules, and (2) a layer of drebrin and f-actin intervenes between the plasma membrane and microtubules in this domain (Fig. 3a,b). We analysed our SR-SIM data with a novel full-width half-maximum (FWHM) Gaussian peak detection algorithm (see Methods) and found clear separation of the respective channels in the SIM images (see the Gaussian peak graphs in Fig. 3a,b). Quantitative measurements in multiple cells from the centre of each Gaussian peak showed the drebrin centre to be ∼170 nm from the plasma membrane centre. The centre of the f-actin or drebrin Gaussian peaks was, on average, between 400 and 430 nm from that of the microtubule Gaussian peak in the proximal leading process.

Although SiR-tubulin labels most of the microtubule cytoskeleton, the growing microtubule plus end represents the putative site of interaction with the cell cortex in well-characterized model systems such as fibroblasts or epithelial cells. The SR-SIM system could not image the microtubule plus end because of its slow imaging speeds and high photobleaching, but LLS microscopy allowed us to probe the dynamic relationship of microtubule plus ends with drebrin E-enriched regions of the proximal leading process. High-speed imaging of CGNs expressing drebrin E-2xKO1 and EB3-2xVenus revealed that EB3-labelled microtubule plus ends are located largely in the central core of the proximal leading process but seldom venture beyond the flanking drebrin (Fig. 3c, Supplementary Video 7), a finding consistent with the overall microtubule architecture defined by our SR-SIM measurement. Taken together, our LLS and SR-SIM imaging reveal a hitherto unappreciated cytoskeletal architecture in the CGN proximal leading process in which the drebrin f-actin–microtubule binding protein translocates to the proximal leading process during the two-stroke motility cycle and a layer of drebrin and f-actin intervenes between the plasma membrane and elements of the microtubule cytoskeleton during CGN migration.

**Drebrin is vital for migration and two-stroke nucleokinesis.** We examined drebrin function by using a well-established *ex vivo*

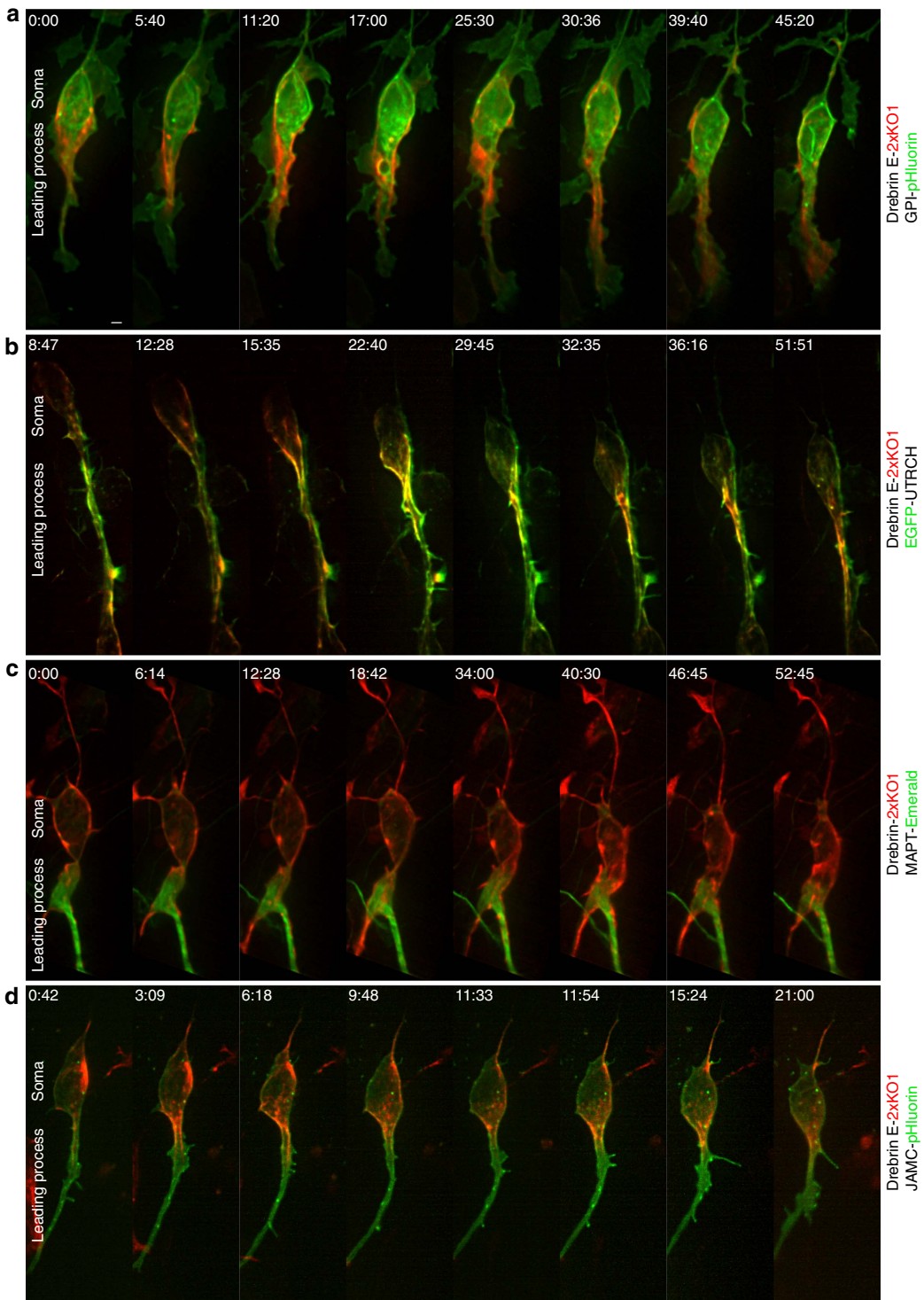

**Figure 2 | LLS microscopy reveals unappreciated relationships between cytoskeleton and plasma membrane in the proximal leading process.** CGNs cultured in conditions that recapitulate radial migration were electroporated with expression vectors encoding the indicated fluorescently labelled live-cell imaging probes. The displayed images are maximum intensity projections of selected time points of cultures imaged via LLS microscopy 18–24 h post transfection. Scale bar, 2 μm. (**a**) Simultaneous imaging of plasma membrane (GPI-pHluorin, green) and drebrin (labelled with drebrin E 2x-KO1, red) reveals that drebrin is submembranous during its anterograde translocation. (**b**) f-Actin (labelled with EGFP-UTRCH, green) accumulates in the leading process before somal translocation. Initially, drebrin (labelled with drebrin E 2x-KO1, red) is localized mostly in the cell body but becomes restricted to the leading process f-actin domain. (**c**) Debrin (labelled with drebrin E 2x-KO1, red) accumulates around microtubules located in a well-dilated proximal leading process (labelled with MAPT-Emerald, green) before somal translocation. Note that drebrin appears to be more cortically restricted than the microtubules. (**d**) Simultaneous imaging of JAM-C adhesions (JAM-C-pHluorin, green) and drebrin (labelled with drebrin E 2x-KO1, red) reveals that a subset of JAM-C adhesions are located in the drebrin contractile domain. Scale bar, 2 μm (**a**). Time stamp = min:sec.

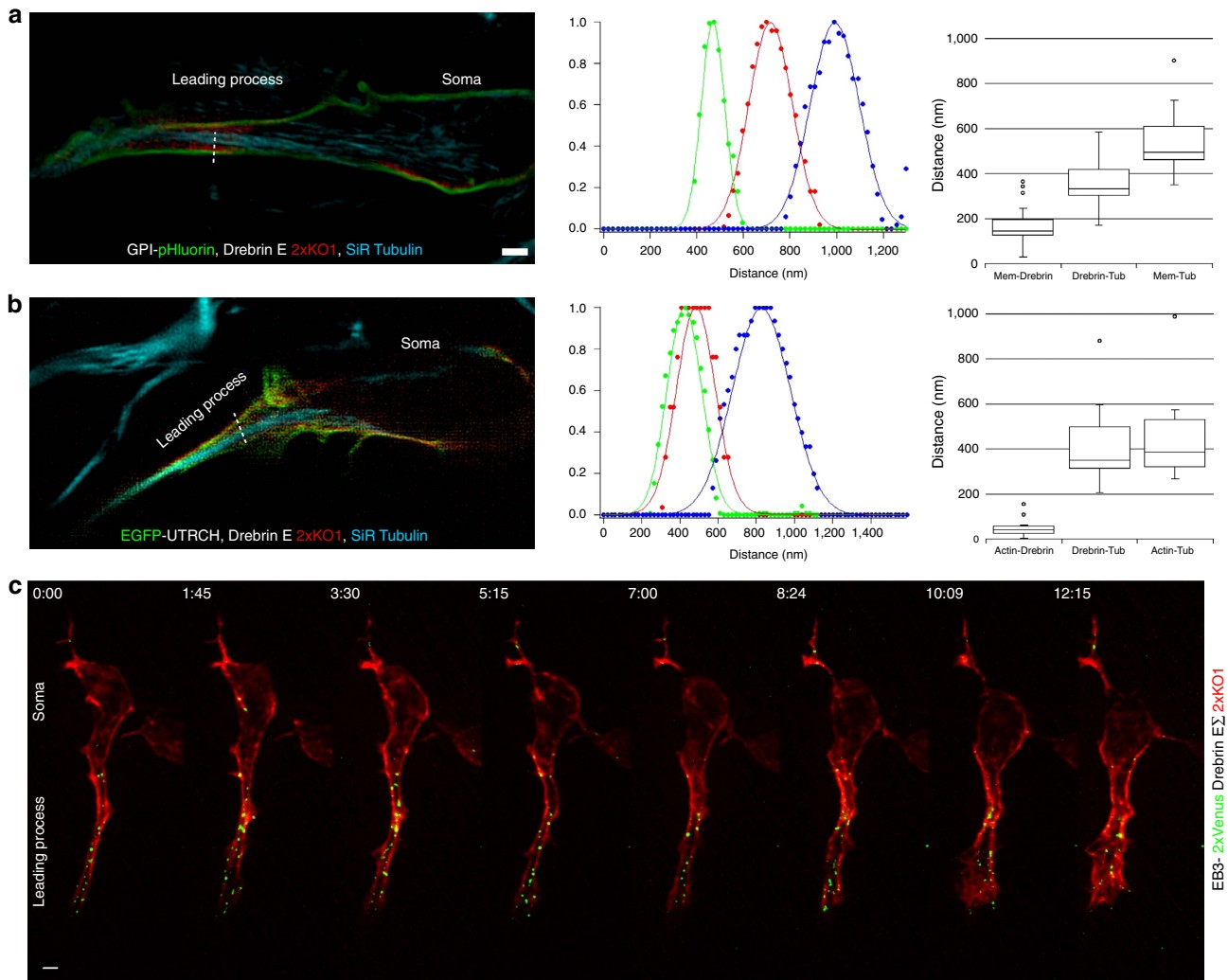

**Figure 3 | SR-SIM reveals that f-actin and drebrin form a cortical collar around microtubules in the proximal leading process. (a,b)** SR-SIM imaging of CGNs with dilated proximal leading processes expressing (**a**) GPI-pHluorin (green), drebrin E 2x-KO1 (red) and SiR-tubulin (blue; $n = 21$ cells analysed) or (**b**) EGFP-UTRCH (green), Drebrin E 2x-KO1 (red) and SiR-tubulin (blue; $n = 18$ cells analysed). Scale bar, 1 µm (**a**). The RGB plot to the right of each representative image shows the well-resolved Gaussian FWHM peaks detected for the line scan in each image (dashed white line). The box plots at far right show the measured distances between the centroid positions of the Gaussian peak for each fluorescent probe. Whiskers on the box plot show maximum and minimum data points, box borders show first and third quartiles and the line in the box show the median. (**c**) Drebrin is an f-actin and +TIP binding protein. Consistent with this interaction, LLS microscopy shows that microtubule +TIPs (labelled with EB3-2x Venus, green) pass through the drebrin E-labelled domain (labelled with drebrin E 2x-KO1, red) in the proximal leading process. Scale bar, 2 µm (**c**). Time stamp = min:sec.

cerebellar slice migration system that recapitulates the migration patterns of developing CGNs *in vivo*, that is, tangential migration near the cerebellar surface followed by radial migration away from the EGL and then across the molecular layer to the IGL[55]. We assayed drebrin function in CGN GZ exit and migration by electroporating a commercially validated drebrin shRNA (Sigma) into the P7 EGL. After 2 days of *ex vivo* culture, CGNs transfected with the control shRNA had migrated into deeper layers of the organotypic slices, whereas *drebrin* silencing decreased CGN migration toward the IGL (Fig. 4a). We next evaluated drebrin function by a dominant-negative approach: overexpression of the drebrin binding region of EB3 (EB3M) titrates drebrin from the plus ends of microtubules, whereas overexpression of the corresponding region of EB1 (EB1M), which cannot bind drebrin, does not[41]. We assayed drebrin plus end-specific functions in CGN GZ exit and migration by electroporating EB3M and EB1M overexpression constructs under the control of the GNP-specific MATH1 enhancer/promoter and the

CGN-specific NeuroD1 promoter. CGNs transfected with control EB1M constructs migrated into deeper layers of the organotypic slices, whereas EB3M overexpression decreased CGN migration toward the IGL in both GNPs and CGNs (Fig. 4a). To examine CGN migration pathway selection, we electroporated P7 EGL with pNeuroD EB1M and EB3M expression constructs and examined the migration of H2B-mCherry-labelled CGNs by long-term time-lapse microscopy followed by quantitative analysis. Consistent with our observations in fixed slices, EB1M-expressing CGNs transitioned to IGL-directed migration after 36 to 48 h of *ex vivo* culture, whereas EB3M-expressing cells underwent no such transition (Fig. 4b, Supplementary Videos 8 and 9). Although the average velocities of EB1M- and EB3M-expressing cells (0.01 µm s$^{-1}$ for EB1M versus 0.009 µm s$^{-1}$ for EB3M) showed little change, detailed analysis of the migration angles relative to the overall vector of movement showed reduced forward migration efficiency of EB3M-expressing CGNs (Fig. 4c). Moreover, mean squared displacement (MSD) calculations

showed these cells spent more of their migration in a random walk (Fig. 4d). Finally, EB3M expression reduced the number of CGNs choosing radial migration angles towards the IGL (Fig. 4e). These results demonstrate that *drebrin* loss of function and dominant-negative inhibition of drebrin plus-end binding

prevent CGN IGL-directed migration, in part by reducing forward movement efficiency and randomizing the migration directions.

Having found that drebrin accumulates in the proximal leading process during the motility cycle and controls IGL-directed

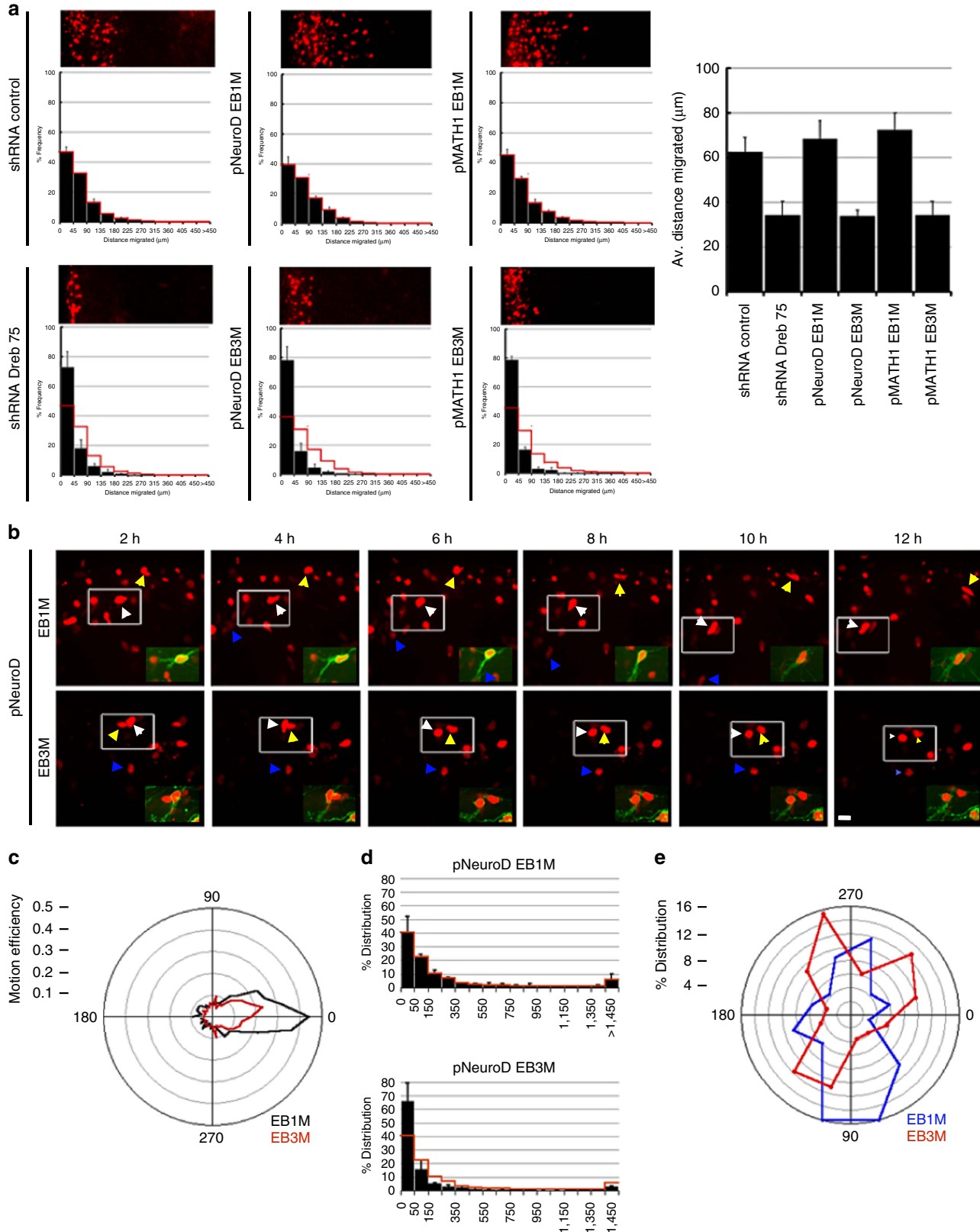

migration, we tested the hypothesis that drebrin regulates two-stroke motility dynamics. We nucleofected expression vectors for the control shRNA, drebrin shRNA, EB1M and EB3M used in our *ex vivo* studies, together with vectors encoding H2B-mCherry/Centrin2-Venus, into purified CGNs and examined centrosome and nuclear migration by time-lapse microscopy of CGN microcultures. *Drebrin* silencing and EB3M overexpression reduced forward motion efficiency and resulted in the accumulation of low MSD, random movements of both the nucleus and centrosome (Fig. 5a–d, Supplementary Videos 10–13) without significantly affecting their average velocities ($\sim 0.01\,\mu\mathrm{m\,s}^{-1}$ for the four experimental conditions). This demonstrates that overall drebrin function and drebrin plus-end interactions are required for polarized centrosome and nuclear movements during the two-stroke movement cycle.

The polarized flow of drebrin during two-stroke motility, its accumulation around leading process microtubules, and the functional requirement of drebrin plus-end interactions for centrosome positioning led us to hypothesize that leading process drebrin couples microtubule movements with the overall migration direction in CGNs. Therefore, we examined how drebrin plus-end interactions affect microtubule positioning dynamics in the proximal leading process by using a novel tubulin photoactivation protocol. We introduced an EB3M expression vector or its control with vectors encoding RFP-UTRCH-ABD (f-actin label) and photoactivatable enhanced green fluorescent protein-α-tubulin (PAEGFP-α-tubulin) into purified CGNs via nucleofection and assayed microtubule positioning by time-lapse confocal microscopy of CGN microcultures. A microtubule region labelled by PAEGFP-α-tubulin was photoactivated in dilated proximal leading processes (visualized by the f-actin RFP-UTRCH-ABD signal) via a brief pulse of 405-nm laser light then imaged for $\sim 20$ min post activation to track the overall cell migration and the microtubule fiduciary mark. In control CGNs, the microtubule fiduciary marks in the proximal leading process advanced in the anterograde direction by an average of $4.72 \pm 0.97\,\mu\mathrm{m}$ during the imaging period, with an average speed of $0.011\,\mu\mathrm{m\,s}^{-1}$, whereas those in the proximal leading process of EB3M-overexpressing cells retreated retrogradely towards the soma by an average of $2.3 \pm 0.8\,\mu\mathrm{m}$, with an average speed of $0.006\,\mu\mathrm{m\,s}^{-1}$ (Fig. 6, $P < 0.01$ for the average advance by Student's *t*-test, Supplementary Videos 14 and 15). *Drebrin* silencing yields similar retrograde movements of $2.34 \pm 0.5\,\mu\mathrm{m}$. This demonstrates that drebrin plus-end interactions are required not only for the polarization of two-stroke motility but also for the appropriate positioning of microtubules within the leading process cytoplasmic dilation.

**The Siah2 ubiquitin ligase antagonizes drebrin function**. We next considered how drebrin activity could be regulated during CGN development. The C terminus of mouse drebrin possesses degron sequences that are conserved binding sites for the Siah2 E3 ubiquitin ligase, which we have characterized as a critical regulator of CGN GZ exit and polarization (see Fig. 7a for a schematic)[55]. Siah degrons, Px[A,R,T]xVxP[59], are rare; computer analyses using ScanProsite, for example, show them to be present in $\sim 440$ proteins in the human proteome. By comparison, the well-characterized anaphase-promoting complex degron xRxxLxx[L,I,VM] appears in over 8,900 proteins in the human proteome. Not only is drebrin expression complementary to that of Siah2 in the P7 cerebellum (Fig. 7b), but sonic hedgehog (Shh) mitogen treatment of cultured CGNs, which blocks CGN differentiation, increased Siah2 expression and decreased drebrin immunoreactivity (Fig. 7c). Overexpression of Siah2—but not of a catalytically inactive Siah2 mutant—diminished the drebrin E-2xVenus fluorescence signal in purified CGNs (Fig. 7d). Moreover, *Siah2* silencing in CGNs led to enhanced drebrin immunoreactivity (Fig. 7e). We further investigated the relationship of Siah2 and drebrin in HEK293 cells. Siah2 expression also reduced drebrin E-2xVenus steady-state protein levels in HEK293 cells in a manner that required the substrate binding domain of Siah2 and VxP degron sequences of drebrin: Siah2 M180K was less efficient than wild-type Siah2 at reducing drebrin levels, while a drebrin E-2xVenus NxN degron mutant was less sensitive to Siah2 expression than wild-type drebrin (Fig. 7f). Interestingly, drebrin E-2xVenus NxN pulls down less ubiquitin than wild-type drebrin, suggesting that the Siah degron is an important determinant of drebrin ubiquination levels (Fig. 7h). CGNs express low levels of a related E3 ligase, called Siah1, which has little functional impact on CGN migration. Siah1 expression in HEK293 cells also reduced drebrin E-2xVenus levels in a manner that required the VxP degron sequences of drebrin and the substrate-binding domain of Siah1 (Fig. 7g).

We used drebrin E-2xVenus NxN to further explore the implications of Siah2 regulation of drebrin in CGNs. Drebrin E-2xVenus NxN fluorescence is more abundant and broadly localized in the CGN leading process than wild-type drebrin (Fig. 7i,j). Moreover, fluorescent recovery after photobleaching (FRAP) showed that drebrin E-2xVenus NxN has a half-life in the cytoplasmic dilation that is two-fold longer than wild type (Fig. 7k), which suggests that drebrin dwell time in the leading process is enhanced while Siah regulation diminished. Taken together, these results suggest the drebrin regulation by Siah E3 ligases can not only tune drebrin expression levels but also localization and residency time in the cytoplasmic dilation.

**Figure 4 | *Ex vivo* analysis of *drebrin* loss of function.** (**a**) Cerebella from P7 mice were electroporated, and slices grown in *ex vivo* culture for 48 h. CGNs were electroporated with a vector encoding H2B-mCherry either alone or in combination with the indicated expression vectors. EB1M and EB3M expression was driven in GNPs using pMath1 and in CGNs using pNeuroD. Each representative image is oriented with the cerebellar slice surface to the left, and the presence of red nuclei in the centre or right of the image indicates cells that have left the GZ. The histograms below each representative image show the binned migration distance distribution for each condition ($n \geq 2,500$ cells analysed for each condition, $\chi^2$ test $P$ value $<0.01$ between each condition and its control), and the graph to the right shows the average migration distances ($P < 0.05$ for all conditions by Student's *t*-test). The migration distance graph for each micrograph is scaled to its accompanying image, thus providing the equivalent of a scale bar. (**b**) Live imaging of *ex vivo* pNeuroD EB1M and EB3M slices confirms migration defects. Representative images in which the cerebellar surface is oriented at the top show the position of H2B-mCherry-labelled nuclei at 2-h intervals. Coloured arrowheads show the relative positions of cells in each frame, and the insets illustrate morphology of cells under each condition. (**c**) Angle analysis of cells tracked in live-cell imaging experiments ($n > 69$ cells for each condition). FDVs were calculated and compared with IDVs for each time. Polar plots representing efficiency show that EB1M-expressing cells had more efficient movements corresponding to their FDV than did EB3M-expressing cells. (**d**) The mean squared displacement (MSD), a Brownian motion measure, was also calculated for each time point of the imaging sequences. The population of EB3M cell movements has more low MSD values, indicating random motion in these cells. (**e**) Radial histogram of the tangential and radial migration angles. Tangential angles are parallel to the 0° to 180° axis (that is, parallel to the slice surface), whereas radial angles are oriented towards 90° (that is, perpendicular to the slice surface). Scale bar, 10 μm. Error bars show ± s.d.

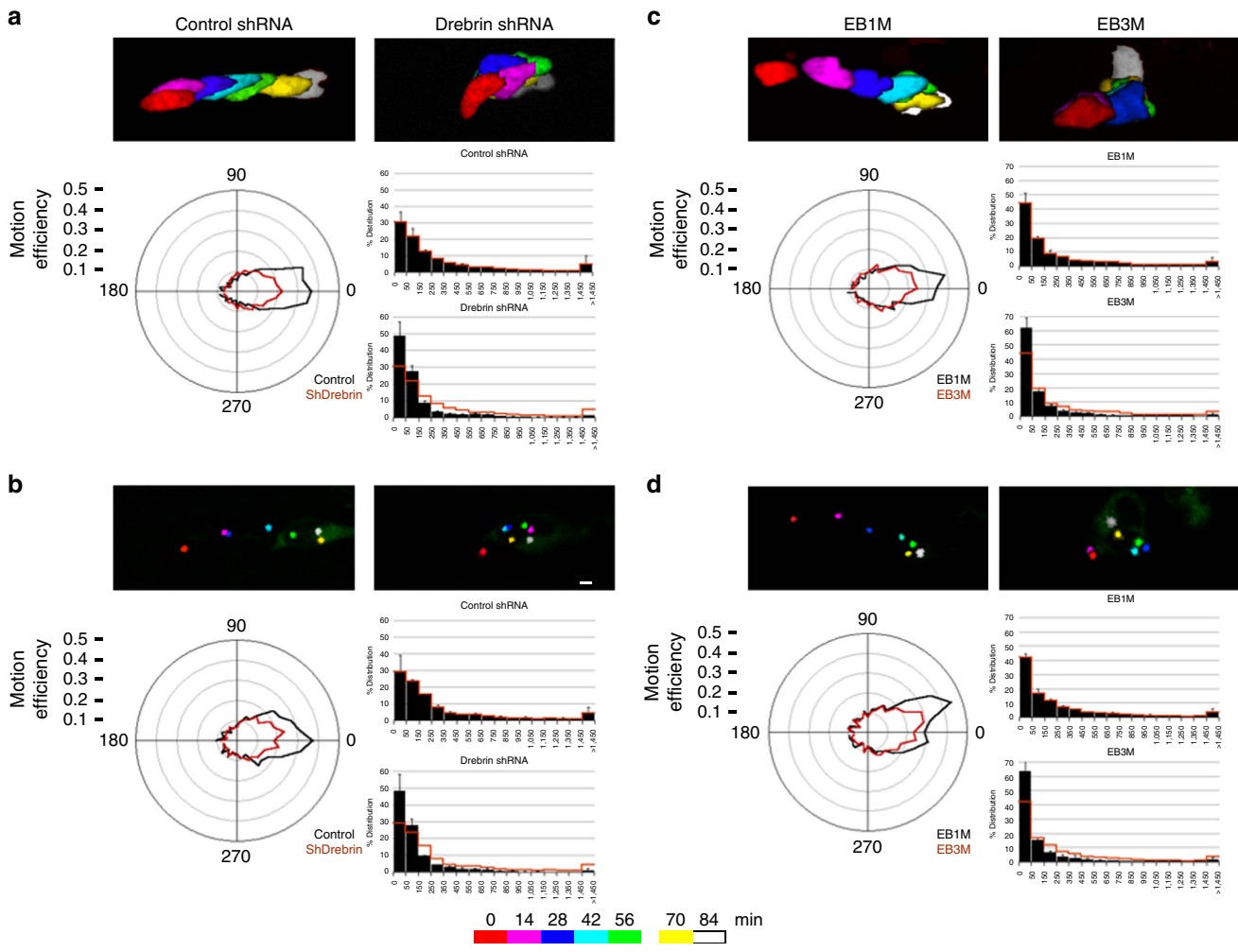

**Figure 5 | Drebrin function is required for directed movement of two-stroke motility *in vitro*.** CGNs were transfected with expression vectors encoding Centrin2-Venus, H2B-mCherry and the indicated experimental constructs. Time-lapse imaging was used to monitor two-stroke nucleokinesis in migrating CGNs in which *drebrin* was silenced (**a,b**) (*n* ≥ 57 cells analysed for each condition and organelle) or when the dominant-negative EB3M was overexpressed (**c,d**) (*n* ≥ 70 cells analysed for each condition and organelle). The multicolour images in **a,c** show nuclear positions for selected time points from representative imaging sequences, whereas **b,d** show centrosome positions (see the colour key for time points). A polar efficiency and an MSD plot are provided for each organelle. *Drebrin* loss of function and dominant-negative inhibition of plus-end binding is accompanied by less efficient somal and centrosome motility relative to calculated FDVs. Scale bars, 10 µm. Error bars show ± s.d.

The Siah2 expression pattern in the cerebellum and Siah2-mediated reduction in drebrin E-2xVenus suggested that Siah2 functionally antagonizes drebrin. To test this hypothesis, we introduced expression vectors for mouse Siah2 or Siah2 plus degron-mutated drebrin (drebrin NxN) into purified CGNs via nucleofection and examined centrosome and nuclear motility by time-lapse microscopy of microcultures. Whereas control CGNs expressing LacZ exhibited typical two-stroke motility in terms of directional persistence along a migration vector and number of high MSD migration movements, Siah2-overexpressing cells were motile, but their centrosome and nuclei showed low directional persistence along migration vectors and an accumulation of low MSD movements, indicative of increased random motility. Interestingly, drebrin NxN expression restored normal leading-process extension (Fig. 8a,b) and somal directional persistence to Siah-expressing neurons (Fig. 8c, Supplementary Videos 16–18), but had no effect on Siah2-induced centrosome motility. This demonstrates that drebrin is required for directed soma movement in the context of *Siah2* gain of function, but other Siah2 targets may be necessary to fully restore centrosome motility.

We also examined the antagonistic relationship between Siah2 and drebrin in *ex vivo* slice preparations. We first tested whether elevated drebrin activity induced IGL-directed migration, given its low expression in GNPs in the EGL. Expression constructs for drebrin NxN and H2B-mCherry were co-electroporated into the P7 EGL. Control CGNs expressing LacZ remained within the EGL after 24 h, whereas CGNs expressing drebrin NxN entered the molecular layer and IGL (Fig. 8d). *Siah2* gain of function has previously been shown to block IGL-directed CGN migration[55]. To test whether drebrin NxN could restore IGL-directed migration in the context of elevated Siah2, expression vectors for mouse Siah2 or Siah2 and drebrin NxN were electroporated into P7 EGL. After 2 days of *ex vivo* culture, control CGNs entered the molecular layer and IGL, whereas Siah2-expressing CGNs remained within the EGL. Drebrin NxN expression in Siah-expressing cells rescued IGL directed migration, much as we observed for somal motility in microcultures *in vitro*. Finally, we tested *Siah-drebrin* epistasis using a loss of function approach where we silenced *Siah2* alone or in combination with *drebrin* silencing. As expected, *Siah2*

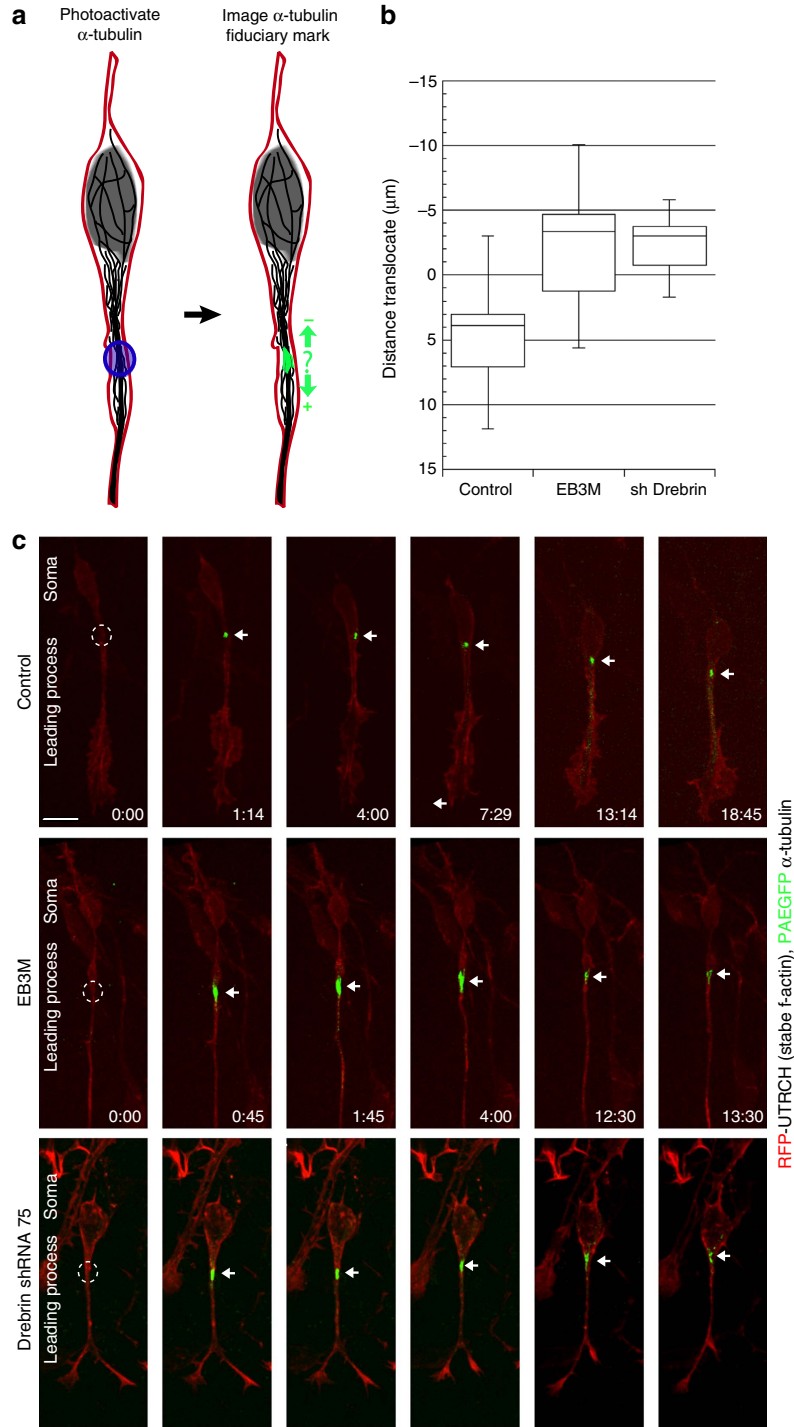

**Figure 6 | Drebrin–microtubule interactions are required for proximal leading process microtubule movement *in vitro*.** CGNs were transfected with expression vectors encoding photoactivatable EGFP-α-tubulin (green), RFP-UTRCH-ABD (f-actin label, red), or the indicated EB3M and drebrin shRNA constructs. (**a**) Schematic of the photoactivation strategy: a 405-nm laser was used to activate a microtubule fiduciary mark within well-dilated CGN proximal leading processes, and time-lapse imaging was used to monitor fiduciary mark movement during two-stroke nucleokinesis. Movements with positive values were towards the distal tip of the leading process (that is, in the direction of migration), whereas negative movement values were towards the cell body. (**b**) Box plot showing the average advance of the fiduciary mark in control, EB3M-expressing or *drebrin*-silenced CGNs. Control marks moved in the direction of migration, whereas the marks in EB3M-expressing cells retreated towards the soma ($n = 15$ for control, 27 for EB3M-expressing and 15 for *drebrin*-silenced cells, $P < 0.01$ for average advance by Student's *t*-test). (**c**) Representative time-lapse frames for each condition: the soma is oriented at the top of each frame, with the leading process oriented down. A circle shows the site of photoactivation, and an arrow highlights the centroid of the fiduciary mark in each frame. Scale bar, 5 μm (**c**). Whiskers on the box plot shows maximum and minimum data points, box borders show first and third quartiles and the line in the box show the median.

silencing induced early GZ exit in cerebellar slices cultured for 24 h. Interestingly, *drebrin* silencing inhibited precocious GZ exit induced by *Siah2* loss of function (Supplementary Fig. 5). At P7 CGNs express low levels of the Siah1 E3 ligase. As expected little change in GZ exit or radial migration is observed when *Siah1* is silenced. Siah1 and drebrin may weakly functionally interact as *drebrin* silencing does not inhibit CGN migration if *Siah1* is silenced. All together, we have demonstrated a hitherto unappreciated functional antagonism between Siah2 and drebrin that suggests regulation of drebrin E-mediated cytoskeletal

interactions is a novel mechanism for controlling the onset of two-stroke motility and IGL-directed migration in maturing CGNs.

## Discussion

Time-lapse microscopy in migrating CGNs shows that drebrin translocates to the proximal leading process before CGN somal translocation, thereby revealing potential sites of microtubule and actomyosin coordination. Lattice light-sheet microscopy and

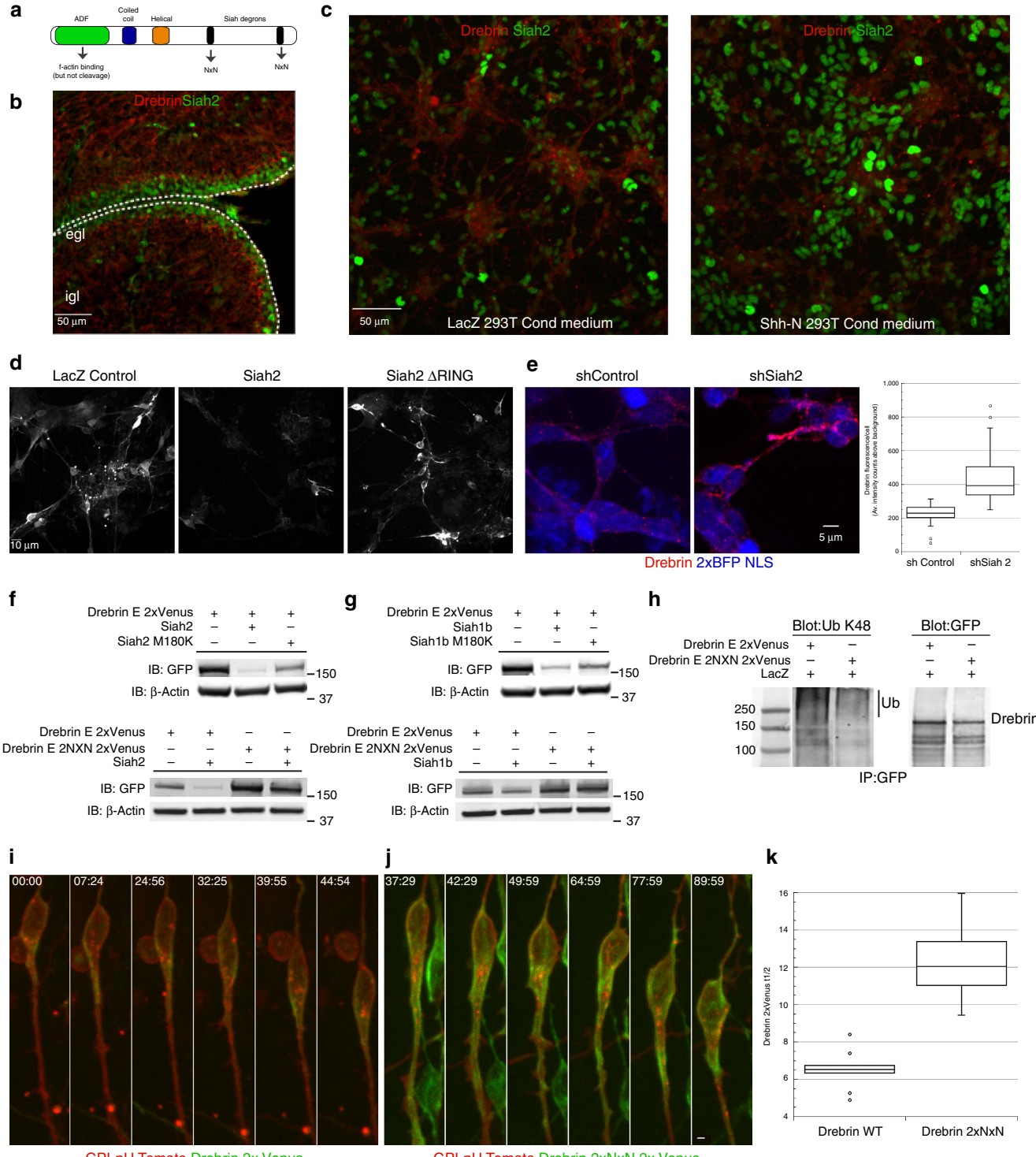

SR-SIM reveal a novel nanoscale architecture of the CGN proximal leading process, with a layer of cortical actin and drebrin intervening between the microtubule cytoskeleton and the plasma membrane. In functional studies, *drebrin* gene silencing and dominant-negative inhibition not only perturb *ex vivo* CGN migration but also randomize centrosome and somal motility dynamics, a functional reporter for actomyosin–microtubule interactions. Finally, we discovered a functional antagonism between drebrin and the Siah2 E3 ubiquitin ligase that restrains CGN GZ exit, neurite extension, and two-stroke motility. Taken together, these findings demonstrate that the proximal leading process represents a region of active coordination between the microtubule and actomyosin cytoskeleton. Moreover, Siah2 regulation of drebrin activity offers insights into how newly differentiated CGNs transition to a two-stroke motility cycle during GZ exit and radial migration.

The neuronal leading process is the locus for the guidance, adhesion and cytoskeletal events that drive neuronal migration throughout the brain[6,8,9,60–62]. Our understanding of the role of leading process microtubule arrays has progressed from investigations into the polarized mechanics of two-stroke motility and the function of cytoplasmic dynein and its cofactors that are mutated in human neuronal migration disorders[19,63,64]. In other cell-biological model systems, microtubule tethering to sub-plasma membrane f-actin or scaffolding molecules associated directly with the plasma membrane is essential in order for microtubule motors, such as cytoplasmic dynein, to exert traction forces on microtubule arrays[31,65,66]. By comparison, how and where microtubules interface with the cell cortex, or f-actin cytoskeleton in migrating neurons is not well understood. Intriguingly, f-actin, f-actin regulatory proteins and f-actin-associated motors are located in the proximal leading process cytoplasmic dilation in an increasing number of neurons and, as with cytoplasmic dynein, the actomyosin cytoskeleton coordinates centrosome positioning and the polarity of two-stroke motility, suggesting there is functional crosstalk in the proximal leading process.

These findings prompted us to examine the relationship of the plasma membrane, the microtubules, f-actin and the drebrin microtubule–actin crosslinking protein in live CGNs. We used a novel combination of imaging probes and super-resolution imaging technologies to gain insights into microtubule-actomyosin cooperation during two-stroke motility. Strikingly, super-resolution imaging revealed an intervening layer of f-actin and drebrin between the plasma membrane and the longitudinal core of the microtubules in the centre of the leading process or EB3-labelled +TIPs. Dynamic live-cell imaging with LLS microscopy and the absence of potential fixation artifacts showed that plasma membrane–microtubule displacement occurs throughout multiple stages of the CGN migration cycle. Although classical electron microscopy studies noted significant displacement between sites of neuron-glial apposition and microtubules[27], our findings define a hitherto unappreciated nanoscale architecture of the proximal leading process in which all the major cytoskeletal components were determined precisely by imaging probes, suggesting that a microtubule interface with the proximal leading process plasma membrane is unlikely. Non-super-resolution imaging of IQGAP[67] and APC[68] function implied that the leading process tip harbours microtubule tethering sites; however, questions regarding the relationship of microtubules to the plasma membrane or f-actin-rich cortex, the nanoscale architecture of these sites in live cells, and the mechanism of microtubule tethering in the proximal leading process were unaddressed. The intervening layer of f-actin and the drebrin microtubule–actin crosslinker indicate that (1) the proximal leading process/cytoplasmic dilation is a novel microtubule tethering site, as suggested by Vallee *et al.*[31] and (2) the microtubule interface in the proximal leading process is probably submembranous f-actin and associated proteins. This cytoskeletal architecture is reminiscent of the relationship between f-actin and microtubules in growth cones, wherein f-actin corrals microtubules, and factors such as drebrin are proposed to enhance the affinity of interactions between these two cytoskeletal polymers[42,69].

Drebrin was an attractive molecular entry point to probe microtubule–actomyosin interactions in migrating neurons as it binds both f-actin and EB3, can recruit microtubules into f-actin-enriched regions in growth cones or dendritic spines, and modulates migration in developing oculomotor neurons or SVZ neuroblasts[41,43–46]. Little is known about the microtubule-coordinating activities of drebrin in migrating neurons, despite studies demonstrating its necessity for efficient tangential and radial migration. Time-lapse analysis of drebrin localization via

**Figure 7 | Siah ubiquitin ligases regulate drebrin protein levels.** (**a**) Drebrin domain structure: the C terminus contains two VxP Siah degrons that can be mutated to NxN to inhibit Siah sensitivity. (**b**) Immunohistochemical examination of drebrin expression in the P7 cerebellum. Drebrin (red) expression is low in Siah2-expressing GNPs (green). Scale bar, 50 µm. (**c**) Immunocytochemical examination of drebrin expression in CGN cultures treated with Shh-N-conditioned medium or LacZ-transfected control. Drebrin (red) expression is lower and Siah2 expression higher (green) in Shh-treated cultures, providing a physiological context for drebrin regulation during differentiation. Scale bar, 50 µm. (**d**) CGNs co-nucleofected with drebrin E 2xVenus and Siah2-myc exhibit lower drebrin signal, but not when Siah2-ΔRING is co-expressed. Scale bar, 10 µm. (**e**) *Siah2* silencing enhances endogenous drebrin expression in cultured CGNs. CGNs were nucleofected with expression vectors, where a control or Siah2 miR30 shRNA was embedded into the 3′ UTR of a 2xBFP NLS cDNA. After drebrin immunostaining, expression levels were measured and displayed in the accompanying graph. Scale bar, 5 µm (Student's *t*-test $P < 0.01$). (**f**) Top: western blotting shows Siah2 expression reduces drebrin protein levels in HEK293 cells, while the Siah2 M180K substrate binding mutant is less efficient at reducing drebrin levels. Bottom: western blotting shows that drebrin E-2xNxN 2x Venus is more abundant at basal level and less sensitive to Siah2 expression than wild-type drebrin E-2x Venus. (**g**) Top: western blotting shows Siah1b expression reduces drebrin protein levels in HEK293 cells, while the Siah2 M180K substrate-binding mutant is less efficient at reducing drebrin levels. Bottom: western blotting shows that drebrin E-2xNxN 2x Venus is more abundant at basal level and less sensitive to Siah1b expression than wild type drebrin E-2x Venus. (**h**) Wild type and 2xNxN drebrin 2xVenus were immunoprecipitated from HEK293 cell extracts and then blotted with antibodies against K48 poly-ubiquitin and EGFP. Wild-type drebrin is significantly more labelled with K48. (**i,j**) Primary CGNs were nucleofected with expression vectors encoding drebrin E-2xVenus wt or the 2xNxN mutant and GPI-pH Tomato. Time-lapse imaging was used to examine the dynamics of wild type and NxN mutant drebrin in migrating CGNs. Note: images are equally scaled to reflect relative abundance drebrin signals. (**i**) Wild-type drebrin (green) is localized as described in Figs 2 and 3. (**j**) Drebrin E 2NxN is more abundant and spreads farther down the leading process, consistent with the increased protein expression observed in HEK293 cells. Time stamp = min:sec. Scale bar, 2 µm. (**k**) FRAP analysis of drebrin 2xVenus wt or 2NxN in the CGN leading process. Drebrin wild type ($n = 9$ cells) or NxN ($n = 10$ cells) was photobleached in regions of interest in the proximal leading process of primary CGNs. The average recovery time of drebrin wild type was 6.5 s and drebrin 2xNxN was 12.25 s, indicating that Siah-insensitive drebrin possesses a longer dwell time in the CGN leading process (Student's *t*-test $P < 0.01$).

LLS microscopy revealed a dynamic transfer of drebrin from the neuronal soma to the proximal leading process before nucleokinesis that is similar to leading-process f-actin flow. *Drebrin* silencing or dominant-negative inhibition of EB3–drebrin interaction

randomized CGN motility in *ex vivo* slice preparations, leading to a failure to transition to radial migration directions.

Mechanistic imaging studies show that *drebrin* loss of function and inhibition of drebrin–EB3 interactions not only randomize

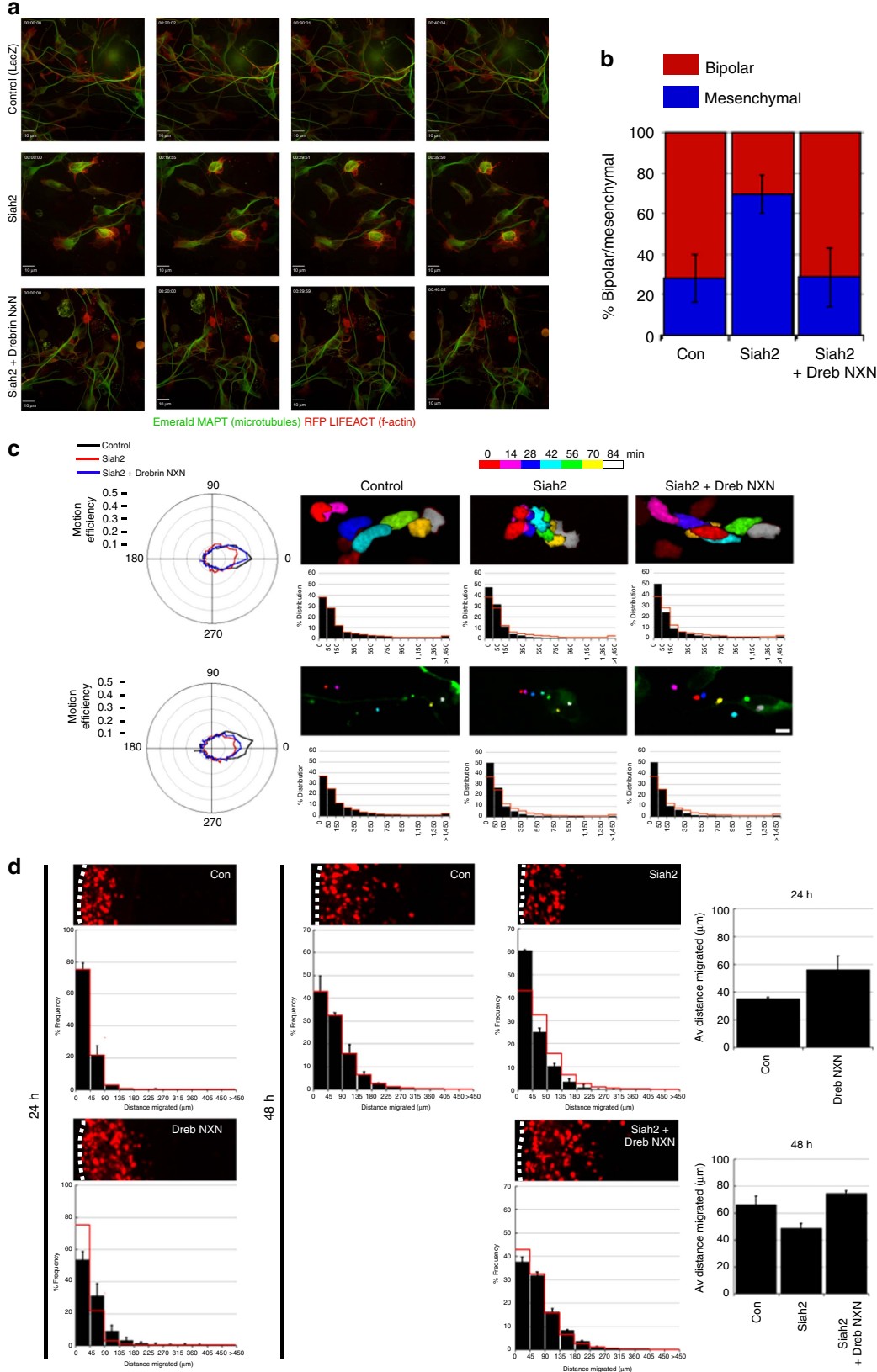

the direction of both phases of the two-stroke motility cycle but also disrupt the positioning of photoactivated fiduciary marks placed on the microtubule cytoskeleton in the proximal leading process while leaving the overall structure of leading process f-actin intact (Fig. 6). Our α-tubulin photoactivation protocol first revealed the predicted translocation of the microtubule arrays in the neuronal leading process. Interestingly, the average velocity of microtubule translocation in wild-type CGNs ($0.011\,\mu m\,s^{-1}$) is nearly identical to that of centrosome translocation and slightly slower than that of f-actin flow. Taken together, these results show that the proximal leading process is a specialized site for microtubule–f-actin interactions that is required to direct the polarity of the two-stroke motility cycle. As f-actin is present before drebrin translocation, the dynamic recruitment of drebrin to the proximal leading process/cytoplasmic dilation suggests that it is an adaptable scaffold coupling f-actin flow to microtubule positioning at key points in the movement cycle. Recent traction force microscopy studies demonstrating contractile units at the leading process tip and proximal leading process have interesting implications for microtubule tethering in the leading process[40]. The present work showing drebrin function in the proximal leading process, together with earlier studies showing IQGAP[67] or APC[68] function at the leading process tip, suggests that the leading process contains distinct microtubule tethering factors near each contractile unit. As drebrin binds to the side of actin filaments, it is well positioned to coordinate microtubule–f-actin interactions in the proximal leading process[70]. We also note in some cases a limited amount drebrin or microtubule plus ends are located in other regions of migrating CGNs. While evidence for actin flow and drebrin function in the leading process is strong, we cannot rule out additional contributions of the drebrin/plus tip system in these other sites. Most relevant is drebrin/EB3 localization to the distal tip of the leading process as drebrin in this region may act as was described in previous studies in the growth cone where it supports process extension[41,47]. It is also possible that random nuclear movement could be due to drebrin function in the cell body and not just randomization of the centrosome. Site-specific drebrin inactivation may be required to answer these detailed mechanistic questions.

Drebrin is regulated through alternative splicing[44] or phosphorylation of its serine 142 residue by CDK5 (ref. 47). Our results demonstrate functional antagonism between drebrin and the Siah2 E3 ubiquitin ligase that controls GZ exit of differentiating CGNs, under the control of the Shh mitogen that regulates GNP proliferation and differentiation. Not only do *drebrin* loss of function and *Siah2* gain of function phenocopy at the level of morphology, migration and two-stroke motility,

and drebrin activity is required for *Siah2* loss of function induced GZ exit, but drebrin NxN, with a mutated Siah degron sequence, also rescues most Siah gain-of-function phenotypes. Moreover, drebrin NxN expression in GNPs, that express low levels of drebrin, induces early GZ exit, demonstrating that drebrin activity is not only essential for IGL directed migration but is also sufficient to initiate the move to a final position. Taken together, these results represent a novel mechanism regulating drebrin function and provide an example of how the cytoskeletal regulatory mechanisms that govern the key processes in two-stroke motility are controlled during neuronal differentiation. In future studies, it will be interesting to define whether proper leading process formation, bipolar morphological maturation or cytoskeletal interactions underlie drebrin's ability to control GZ exit and the transition to radial migration.

## Methods

**Animals.** All mice in this study were housed, bred and used according to the procedures and guidelines approved by Institutional Animal Care and Use Committee at St Jude Children's Research Hospital (protocol number = 483).

**Cerebellar immunohistochemistry.** Postnatal brains collected at p7 were fixed by overnight immersion in 4% paraformaldehyde at 4 °C followed by cryoprotection in PBS containing 30% sucrose. Histologic sagittal sections were cut at 16-µm thickness on a cryostat and preblocked for 1 h in PBS with 0.1% Triton X-100 and 10% normal donkey serum. Sections were incubated overnight at 4 °C with the primary antibodies, followed by incubation with the appropriate Alexa-labelled secondary antibody (Invitrogen) at 1:1,000 for 1 h before mounting on slides. Antibodies used were mouse α-drebrin A/E (Clone 2E11, Sigma Catalogue Number SAB1402168, 1:5,000 dil), rabbit α-drebrin A/E (Abcam Catalogue Number ab11068, 1:5,000 dil), rabbit drebrin A (DAS2, Immuno-Biological Laboratories Co., Ltd, Catalogue Number #28023, 1:100 dil), rabbit α-myosin IIB (Covance Catalogue Number 909901, 1:100 dil), mouse α-alpha tubulin (Tub2.1, Sigma Catalogue Number T4026, 1:1,000 dil), and α-phospho drebrin Ser142 (clone 3C14, EMD Milipore Catalogue Number MABN833, 1:100 dil).

**Preparation and nucleofection of CGNs or *ex vivo* brain slices.** CGNs were prepared according to the established protocols[35]. Briefly, cerebella were dissected from the brains of P7 C57BL/6 mice, the pia was peeled away, and the cerebellar tissue was treated with trypsin and triturated using a fine-bore fire-polished Pasteur pipettes. A single-cell suspension in CMF-PBS was layered onto a 60–35% Percoll gradient and separated by centrifugation. The cellular fraction at the interphase was isolated (it routinely contained 95% CGNs and 5% glia) and transfected by nucleofection. For imaging experiments, CGNs were nucleofected with 1–20 µg of pCIG2 expression vector encoding fluorescence-labelled cytoskeletal proteins by using the Amaxa Mouse Neuron kit with the A030 or O-005 program on the Amaxa Nucleofector II system (Lonza). Nucleofected CGNs were plated in glass-bottomed movie dishes (MatTek Corporation) coated with poly-L-ornithine and Matrigel at a low concentration to facilitate the attachment of neurons to glial processes.

*Ex vivo* brains slices were prepared and imaged imaged with a Marianas spinning disk confocal microscope[55], with the exception that the slices were processed so as to retain an agarose cushion to avoid tissue deformation.

**Figure 8 | Siah2 antagonizes drebrin function.** (**a**) CGNs in culture expressed Emerald MAPT and RFP LIFEACT to label the microtubule and actin cytoskeleton. Time-lapse imaging shows that Siah2-insensitive drebrin NxN rescues a Siah2 gain-of-function phenotype. Top row: Control CGNs have long neurites. Middle row: *Siah2* gain of function inhibits CGN neurite extension, induces a radial microtubule cytoskeleton and locks CGNs in a mesenchymal morphology. Bottom row: CGNs expressing drebrin NxN and Siah2 have neurites and microtubule cytoskeleton similar to controls. Scale bar, 10 µm. (**b**) Quantitation of imaging sequences shown in **a** ($n \geq 337$ cells analysed for each condition, $P < 0.01$ by Student's *t*-test for differences between the Siah2 condition and the control or Siah2 + drebrin NxN). (**c**) CGNs were transfected with expression vectors encoding Centrin2-Venus, H2B-mCherry, and the indicated constructs. Time-lapse imaging was used to monitor two-stroke nucleokinesis in migrating CGNs in which Siah2 was overexpressed or rescued with drebrin NxN ($n \geq 107$ cells analysed for each condition). The multicolour images show nuclear/centrosome positions for selected time points from representative imaging sequences, and the polar Efficiency and MSD plots display movement characteristics. *Siah2* gain of function randomized nuclear and centrosome positions, but only nuclear position is rescued by drebrin NxN. Scale bar, 5 µm. (**d**) Cerebella from P7 mice were electroporated and slices grown in *ex vivo* culture for 48 h. The cells were electroporated with a vector encoding H2B-mCherry either alone or in combination with the indicated expression vectors. Each representative image is oriented with the cerebellar slice surface to the left; the red nuclei in the centre or right of the image indicate cells that have left the GZ. The histograms below each representative image show the binned migration distance distribution for each condition ($n \geq 3,893$ cells analysed for each condition, $P < 0.01$ by $\chi^2$ test for each condition and its control). The graphs to the right show the average migration distances ($P < 0.05$ by Student's *t*-test for all conditions). The migration distance graph for each micrograph is scaled to its accompanying image, providing the equivalent of a scale bar for each image. Error bars show $\pm$ s.d.

**Plasmid vectors.** All cDNAs encoding fluorescent fusion proteins were subcloned into the pCIG2 expression vector or into vectors harbouring the NeuroD and Math1 promoter/enhancer regions. pCIG2 H2B-mCherry, pCIG2 UTRCH-ABD-RFP and pCIG2 JAM-C-pHluorin were described previously. Gero Miesenboeck provided the pHluorin cDNA, Atsushi Miyawaki provided the Venus cDNA. The Arl13b-Venus, drebrin E-2xKO2, drebrin E-2xVenus, EB3M, EB1M, GPI-pHluorin, PA-EGFP-α-tubulin, Eb3 2xVenus and 2xVenus cytoplasmic dynein cDNAs were commercially synthesized and subcloned into pCIG2 by GenScript (Piscataway, NJ).

**Standard imaging and analysis.** CGN cultures (Figs 1 and 6–8) and *ex vivo* slices were imaged with a Marianas Spinning Disk confocal microscope (Intelligent Imaging Innovations) consisting of a Zeiss Axio Observer microscope equipped with × 40/1.0 numerical aperture (NA) Plan-Apochromat (oil immersion), × 63/1.4 NA Plan-Apochromat (oil immersion) and × 40 C-Apochromat 1.2 W Corr M27 (WD = 0.14–0.28 mm) objectives for high-magnification *ex vivo* brain-slice imaging. An Ultraview CSUX1 confocal head with 440–514 nm or 488–561 nm excitation filters and an ImageEM-intensified CCD camera (Hamamatsu) were used for high-resolution imaging. Z-stacks (10 μm thick, 13 sections per stack) were collected over the duration and time intervals indicated. Video recordings were captured using Slidebook software (Intelligent Imaging innovations), and motion analysis was obtained using a custom-designed MATLAB centrosome and nuclear tracking tool designed by the Imaging, Signals and Machine Learning Group at Oak Ridge National Laboratory and mean squared displacement (MSD) was extracted in Slidebook. For angle analyses, final displacement vectors (FDVs) were calculated from the initial and final position in each time series as well as an instant displacement vector (IDV) for each time point. For each time point, an angle between FDV and IDV was calculated, and the magnitude of the IDV scored for that angle. Each pair (angle/magnitude) were then binned by 10° increments. For each bin, a probability ($n/n$Total), average speed (average magnitude) and the product (motion efficiency = probability × average speed (pixels/time)) was calculated. The polar plots in Figs 4c, 5a–c and 7e show the distribution of Efficiency. The 3D cell segmentations describing drebrin 2xKO1 localization in migrating CGNs were carried out using an adaptive kymograph tool[26]. Cells were only included in the study if they showed good health throughout the length of the movie and the organelles of interest were visible over at least five consecutive frames.

**Lattice light-sheet microscopy.** Images were acquired on a lattice light-sheet microscope built by Intelligent Imaging Innovations, Inc. The specialized optics of this instrument have been described by Chen *et al.*[52]. The optics used for the experiments herein included the following: 560- and 488-nm laser lines, with maximum power of 500 and 300 mW, respectively, and a quad-band emission filter to resolve spectrally the imaged channels.

The LLSM was aligned each morning, allowing at least 4 h for thermal equilibration after the heating apparatus was activated. Briefly, the light path was aligned using Hibernate-low fluorescence medium (Brainbits) containing a fluorescent dye. After rinsing the sample chamber to remove the dye and exchanging the medium, point-spread functions (total and XZ) and Z-Galvo offsets were derived from the imaging of TetraSpeck microspheres.

All the cell samples were imaged in Hibernate low-fluorescence medium supplemented with 10% heat-inactivated horse serum, to which the spherical aberration correction parameter of the objective was adjusted. Cells were seeded on 5-mm cover slips and mounted in custom-fabricated sample holders for imaging. Images were acquired using the LLSM software package in dual colour mode, with both colours being captured at each *z* position.

The acquired images were background subtracted and deskewed using Slidebook software. Lucy–Richardson deconvolution was performed in Slidebook using a MATLAB bridge based on point-spread functions derived from imaging TetraSpecks before imaging the cells. Series export was performed using the standard Slidebook package.

**Super-resolution structured illumination microscopy.** Images were acquired with a commercial Elyra PS.1 microscope (Carl Zeiss, Inc.) equipped with a Plan-Apochromat × 100/1.46 Oil DIC objective and 405, 488, 561 and 642 nm laser lines. Live-cell triple labelling was achieved by staining CGN cultures that had been nucleofected with the indicated genetically encoded red/green fluorescent protein pairs with SiR tubulin dye according to the manufacturer's instructions (Spirochrome). CGNs with dilated proximal leading processes and well-defined triple labelling were selected for imaging and analysis. After acquisition, the images were processed for structured illumination and deconvolution via Zen Black edition software, and the relationship between the labels was analysed in single optical sections in which each label was well defined. The intensity profile of a line crossing the proximal leading process was generated in Zen 2012 Black software. The line profiles were then loaded into Igor Pro software (Wavemetrics Inc., Portland, OR) to normalize and fit them with a Gaussian equation:

$$y = y_0 + A \exp\left(-\frac{(x-x_0)^2}{2\sigma^2}\right),$$

where $y_0$ is the offset, $A$ is the amplitude, $x_0$ is the centre of the Gaussian peak, and $\sigma$ is the s.d.

The fitted results $x_0$ and $\sigma$ were used to calculate centre-to-centre distances between profiles: red-to-green, red-to-blue and green-to-blue. The centre-to-centre distance was simply an absolute value of the difference in the Gaussian peak centres. The edge-to-edge distance (in this example for red-to-green) was calculated as follows:

$$\left(x_0^{\text{Red}} - \frac{\text{FWHM}^{\text{Red}}}{2}\right) - \left(x_0^{\text{Green}} - \frac{\text{FWHM}^{\text{Green}}}{2}\right),$$

where FWHM is the full width at half maximum of the Gaussian peak, which was calculated using the Gaussian s.d. as follows:

$$\text{FWHM} = 2\sqrt{2\ln 2}\sigma.$$

A negative value for the edge-to-edge distance indicates overlapping peaks.

**Western blot.** Hek293 cells were lipofected (Lipofectamin 2000, Thermofisher) one day before being harvested. Cells were lysed using Lysis/Binding/Wash Buffer (Cell Signaling Technologies) and reduced using Laemmeli buffer 2 × (Tris-HCl pH6.8 125 mM, Glycerol 20%, SDS 4%, Bromophenol Blue 0.02%). Samples were loaded using the iBlot system (Thermofisher) and immunoblotted using GFP (Rabbit 1:5,000 dil, A11122, Life Technologies) and β-actin (Mouse monoclonal 1:15,000 dil, A2228, Sigma) or drebrin antibodies mentioned earlier and revealed with Li-Cor secondary antibodies (926-65010 and 925-32211, LI-COR). See Supplementary Fig. 6 for selected uncropped blots displayed in Fig. 7.

**Statistical analysis.** All data were expressed as the mean ± s.d. The Student's *t*-test was used for comparing two groups with the level of statistical significance set at $P < 0.05$–.01 unless otherwise specified. In migration rescue assays, if rescuing conditions resulted in a $\chi^2$-test $P$ value > 0.8 when compared with controls, and *t*-test $P$ value < 0.01 when compared control migration distance. Absolute numbers of measurements are reported in figure legends and most studies utilized three biological replicates.

**Data availability.** The data that support the findings of this study are available from the corresponding author upon reasonable request.

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

## Acknowledgements

The Lattice Light Sheet Microscope referenced in this research was used under license from Howard Hughes Medical Institute, Janelia Research Campus. We thank Drs Boyd Butler, Owen Richards, Glen Redford, Karl Kilborn and Colin Monks from 3i for their efforts implementing LLS, James McMurry, Bill Pappas and Andrew Pappas from the St Jude Information Science department for computational support of LLS and Cell and Tissue Imaging Core of St Jude Children's Research Hospital for assistance implementing SR-SIM imaging. Keith A. Laycock, PhD, ELS edited the manuscript. The Solecki Laboratory is funded by the American Lebanese Syrian Associated Charities (ALSAC), by grant #1-FY12-455 from the March of Dimes and by grant 1R01NS066936 from the National Institute Of Neurological Disorders (NINDS). The content is solely the responsibility of the authors and does not necessarily represent the official views of the NINDS or the NIH. The Gordon–Weeks laboratory is supported by the Biotechnology and Biological Sciences Research Council (BBSRC). This manuscript has been authored by UT-Battelle, LLC under Contract No. DE-AC05-00OR22725 with the US Department of Energy.

## Author contributions

N.T. carried out *ex vivo* studies, Siah/drebrin epistasis analyses and prepared many figures. D.R.S. spearheaded LLS microscopy implementation for the study carried out drebrin LLS imaging. B.C. carried out all *in vitro* two-stroke nucleokinesis imaging and used ORNL developed algorithms to analyse migration experiment in Figs 4b–e, 5 and 8a–c. D.H. performed all immunostaining and prepared CGN cultures throughout the study. C.L. designed and performed all western blotting or immunoprecipitation experiments. J.S.R. performed the experiments in Supplementary Fig. 1 and 2, designed and performed the first rounds of tubulin photoactivation experiments and performed proof of principle experiments examining myosin II's role in drebrin translocation. J.T. wrote the Igor Pro analysis algorithm for SR-SIM analysis and trained D.J.S. in the use of the Zeiss Elrya microscope. J.K. designed the adaptive kymograph and centrosome/soma tracking algorithms used throughout the study. P.R.G.-W. participated in conceptual study design, provided many drebrin constructs or antibodies used in the study. D.J.S. conceived of the study, participated in its design and coordination and performed all SR-SIM imaging studies. All authors drafted or edited the manuscript.

## Additional information

**Competing financial interests:** The authors declare no competing financial interests.

