## [Peer Review File · Nature Communications]

Reviewers' Comments:

Reviewer #1 (Remarks to the Author)

This is an interesting and important study to elucidate the role of drebrin in the neuronal migration during development. The authors showed by LLS and SR-SIM imaging that drebrin translocates to the proximal leading process during the two-stroke motility cycle. Then they demonstrate using dominant-negative inhibition by EB3M that specific binding of drebrin and EB3 play a role in cell migration. Thirdly they show Siah2 negatively regulate drebrin activity and ubiquitin proteasome is required for this Siah2 regulation. Finally the author suggested that drebrin-mediated cytoskeletal interactions is a novel mechanism for controlling the onset of two-stroke motility and IGL directed migration in maturing CGNs.

This referee think that the methods are fine and the conclusion is based on the results.

However there is one thing that the author should clarify. Drebrin has two major isoforms in mouse, drebrin E and drebrin A and minor isoform s-drebrin A (Jin et al. 2002), and the drebrin in migrating neurons is drebrin E (Song et al. 2008). However, in this manuscript there is no mentioning about the difference of isoforms. Is the construct drebrin-2xKO1 is made with drebrin E cDNA or drebrin A cDNA?

And how specific are the antibodies used in this study? The authors should show that the drebrin image acquired in this experiment is drebrin E but not A using drebrin A specific antibody.

In the discussion, please cite the following papers (Shirao and Obata, Dev. Brain Res.29: 233-244, 1986) and (Shirao et al., Neurosci. Res. 13: S106-S111, 1990) about the drebrin expression pattern in developing cerebellum and (Tanaka et al., Neuroscience 97:727-734, 2000) about drebrin expression in migrating granule cells.

Minor points:

P15L1; "debrin1" must be "drebrin".

P17L19; authors mentioned about neurite extension and indeed they showed the ratio of bipolar cells in Fig.7d, however there was no data about neurite length. Are there any difference of neurite length among three types of cells?

P20L6; "drebrin1" should be "drebrin".

P20L11~12; Please cite a reference for "...demonstrating its necessity for efficient tangential and radial migration."

P24L9; "drebrin 2xKO2" must be "drebrin 2xKO1".

P35L17; "debrin1" must be "drebrin".

Fig.1 h; said "Displayed in Panel A", but where is the panel A?

Fig.3 a and b; in the box plots, what do the error bars represent? And what do the small open circles stand for?

Fig.6 b; error bars?

Fig.7 a; the resolution is too bad and cannot be recognized.

Fig. 7 c; should indicate the quantitative data of the blotting.

Reviewer #2 (Remarks to the Author)

The migration of cerebellar granule neurons involves "two strokes" nucleokinesis cycles whose molecular regulation remains unclear. By using a combination of new imaging probes and super-resolution microscopy, the authors gain insight into the dynamic molecular cooperation existing between the microtubules and the actomyosin that underlie CGNs migration. The present work describes a dynamic relocation of active drebrin between the MTs and the plasma membrane in the proximal part of the leading process before nucleokinesis. This protein can bind f-actin and EB3 and is known to control MT movements to actomyosin in the growth cone. The functional assays performed in this study suggest the requirement of drebrin in the proximal region of the leading process for the dynamic coupling between microtubule (MT) and actomyosin during CGN migration. Moreover, the authors add another layer of regulation by suggesting the existence of a dynamic regulation of drebrin degradation by its E3 ligase Siah2 during migration.

Majors

> By combining cutting-edge imaging technologies, this study convincingly shows the existence of a dynamic regulation of MT-actomyosin coupling by drebrin in migrating CGNs. However, it is not clear how drebrin is translocated forward in the proximal leading process before nucleokinesis.

> Drebrin has to be phosphorylated (active configuration) to interact with both actomyosin and MTs. The figure 1, Panel F shows overlap of drebrin with its phosphorylated form in the entire cell (including the distal leading process) with a stronger phospho staining around the nucleus. The authors should provide images of stainings performed with both antibodies in cells undergoing different phases of the two-stroke process. This would help clarifying whether drebrin translocates into the initial segment of the leading process after phosphorylation or if phosphorylation is required/dispensable for its translocation. The authors should also provide split images of the two stainings for a better comparison between the localization and abundance of phospho-drebrin/drebrin.

> In figure 2, the authors show time lapse sequences of cells electroporated with drebrin-2xKO1 and three other markers: for actin, MTs and adhesion sites. Why is the localization of drebrin-2xKO1 so different across panels (e.g. Fig. 2a and 2d drebrin-2KO1 is present in the trailing process but not in cells depicted in Fig. 2b and 2c; in Fig 2a drebrin-2KO1 is also localized at the distal leading process, which is not obvious in cells depicted in other panels of Fig. 2)? Can the authors comment on the heterogeneity of the localization of drebrin-2KO1 across the population of CGNs?

> Tracking of photoactivated labeled alpha-tubulin suggests a role for drebrin in the dynamic repositioning of MTs in the proximal leading process. However, this experiment has been done with the dn construct (EB3M) that titrates drebrin from MTs but that do not directly prevent its interaction with actomyosin (and per se may unbalanced the existing tight feedback regulation of actomyosin on MTs via other putative mechanisms). Can that author demonstrate that the MT phenotype is recapitulated with drebrin shRNA??

> Drebrin interacts with MTs at the +/-TIP ends. This interaction is suggested to be particularly important in the proximal region of the leading process for the control of migration. However, EB3 and drebrin are also colocalized in more distal parts of the leading process (Figure 3C), in the cell body or at the rear of the nucleus. What is the contribution of drebrin-MTs interaction in these regions to the migration process?

Minors

> In the figure 1, the panel in h shows the sequence in h not A as mentioned on the figure.

> The figure 2a should be contrasted to appreciate the specific location of the stainings.

>In the figure 7c: the addition of the catalytic dead Siah2 deltaRing does seem to reduced expression of Drebrin-Venus (lane 1, 3rd well). Could the author comments?

Reviewer #3 (Remarks to the Author)

This manuscript used several elegant microscopy techniques to investigate the dynamic localization and function of drebrin in migrating CGNs. It demonstrates that drebrin functions in the proximal leading process microtubule positioning, microtubule-actomyosin coupling, two-stroke nucleokinesis, and directed migration of CGNs. The authors also suggest that the drebrin function is antagonized by the E3 ubiquitin ligase Siah2. The imaging study of the drebrin localization and function is impressive and well performed.

However, the antagonism of drebrin function by Siah2 is a major weakness of this study due to the approach of Siah2 overexpresssion. Siah2 is very difficult to be detected in most cells possibly due to the self-ubiquitination/-degradation. The overexpression of Siah2 usually leads to the non-physiological high levels of Siah2 and artificial degradation of many proteins.

1. It will be important to knock down the Siah2 in the CGNs and examine the effect on the drebrin ubiquitination, protein level, stability, localization and function.
2. Is the level, activity or localization of endogenous Siah2 altered during the CGN migration, and under what physiological condition that endogenous Siah2 degrades drebrin and regulates the CNG migration?
3. Siah2 knockout mice have no apparent phenotype in the neuron system. It will be interesting to see whether Siah1 expresses in CGNs, and whether both Siah1 and Siah2 are required for the regulation of drebrin and CNG migration.
4. The biochemical analysis of Siah2 overexpression on drebrin degradation is insufficient. The western blot in Figure 7c shows that the catalytic-inactive Siah2 ring mutant seems to also reduce the level of GFP-tagged drebrin. The authors need to provide more convincing evidence that Siah2 overexpression indeed changes the ubiquitination and stability of drebrin, and the drebrin NxN mutant shows a reduced interaction with Siah2.
5. For the functional analysis of Siah2 overexpression or Siah2 overexpression plus Drebrin NxN mutant in Figure7, it will be important to show the western blot using the drebrin antibody, to prove whether Siah2 overexpression decreases the level of endogenous drebrin and whether the expression of the drebrin NxN mutant restores the loss of endogenous drebrin.

Reviewer 1

Point 1: However there is one thing that the author should clarify. Drebrin has two major isoforms in mouse, drebrin E and drebrin A and minor isoform s-drebrin A (Jin et al. 2002), and the drebrin in migrating neurons is drebrin E (Song et al. 2008). However, in this manuscript there is no mentioning about the difference of isoforms. Is the construct drebrin-2xKO1 is made with drebrin E cDNA or drebrin A cDNA?

Response: We used the drebrin E cDNA for these studies and have noted this in the text.

Point 2: And how specific are the antibodies used in this study? The authors should show that the drebrin image acquired in this experiment is drebrin E but not A using drebrin A specific antibody.

Response: The reviewer brings up an important point. The main antibody for our initial submission recognizes both drebrin A and E. We have extended the expression analysis by also staining with a drebrin A antibody as suggested. As can be seen in Figure 1d, there is almost no drebrin A immunoreactivity in the P7 CGNs and perhaps a small amount in the maturing Purkinje cells. Thus, we can be confident that drebrin E is the isoform relevant at this time in cerebellar development.

Point 3: In the discussion, please cite the following papers (Shirao and Obata, Dev. Brain Res.29: 233-244, 1986) and (Shirao et al., Neurosci. Res. 13: S106-S111, 1990) about the drebrin expression pattern in developing cerebellum and (Tanaka et al., Neuroscience 97:727-734, 2000) about drebrin expression in migrating granule cells.

Response: the suggested citations have been added.

Reviewer 2

Point 1: It is not clear how drebrin is translocated forward in the proximal leading process before nucleokinesis.

Response: The reviewer raises an important question. Previous work from our laboratory documented a myosin ii driven f-actin flow in the leading process. This flow translocates from the soma-cytoplasmic dilation to the distal tip of the leading process. We treated migrating CGNs expressing drebrin E-2x Venus with 25 μ M Blebbistatin and observed that drebrin dynamics immediately halted after inhibition of myosin ii motor activity. Thus, leading process myosin ii motor activity is one potential motor system responsible for drebrin translocation. These data are included in Supplementary Figure 4.

Point 2: Drebrin has to be phosphorylated (active configuration) to interact with both actomyosin and MTs. The figure 1, Panel F shows overlap of drebrin with its phosphorylated form in the the entire cell (including the distal leading process) with a stronger phospho staining around the nucleus. The authors should provide images of stainings performed with both antibodies in cells undergoing different phases of the two-stroke process. This would help clarifying whether drebrin translocates into the initial segment of the leading process after phosphorylation or if phosphorylation is required/dispensable for its translocation. The authors should also provide split images of the two stainings for a better comparison between the localization and abundance of phospho-drebrin/drebrin.

Response: We performed a more detailed staining with drebrin and phospho Ser142 drebrin using Centrin2-Venus labeled centrosomes to stage two-stroke motility (See Supplementary Figure 3). Prior to centrosome polarization, phospho Ser142 is diffuse through the CGN cell body. Ser142 is then seen at higher level nears the site of initial centrosome polarization. When a defined dilation is present phospho Ser142 is evident in the proximal leading process, suggesting the drebrin phosphorylation cycle may be related to the forward flow we observed with labeled drebrin in live CGNs.

Point 3: Can the authors comment on the heterogeneity of the localization of drebrin-2KO1 across the population of CGNs?

Response: Some of the variability in localization is due to the fact the cultured CGNs are not homogeneous in or leading/trailing process length or synchronized in regards to their migration status, contributing to the appearance of heterogeneity. Another factor for localization heterogeneity is CGNs migrating along glia in culture stochastically change migration direction where components like the centrosome and leading process cytoskeletal material translocate from leading process to trailing process. The example shown in Figure 2C underwent such a direction change later in the imaging sequence. We've added text to the Results section regarding this issue.

Point 4: Can that author demonstrate that the MT phenotype is recapitulated with drebrin shRNA?

Response: We have repeated photoactivation experiments with the drebrin shRNA used throughout the study. As we saw with EB3M over-expression, drebrin silencing also caused the retreat of photoactivated microtubule fiduciary marks. The additional data is available in Figure 6.

Point 5: Drebrin interacts with MTs at the +-TIP ends. This interaction is suggested to be particularly important in the proximal region of the leading process for the control of migration. However, EB3 and drebrin are also colocalized in more distal parts of the leading process (Figure 3C), in the cell body or at the rear of the nucleus. What is the contribution of drebrin-MTs interaction in these regions to the migration process?

The reviewer brings up an important point: in some cases a limited amount drebrin or the plus ends of microtubules are located in other regions of migrating CGNs. To address the reviewer's concern we have added additional text to the discussion. Our previous results demonstrate that actin exchange between the soma and proximal leading process heavily influences centrosome/nuclear positioning. The exchange of drebrin between soma and leading process in addition to the perturbations of dilation microtubule dynamics/centrosome positioning by drebrin loss of function strongly suggest a role for drebrin in this region. To present a more balanced argument we added text to the manuscript saying we can't formally rule out additional contributions of drebrin/plus tip system in other regions of migrating neurons. Most relevant is drebrin/eb3 localization to the distal tip of the leading process. We now note that drebrin in this region may act much like what has been described for previous studies in the growth cone, where drebrin supports process extension. We also raise the possibility that random nuclear movement could be due to drebrin function in the cell body and not just randomization of the centrosome. Site-specific drebrin inactivation may be required to answer these detailed mechanistic questions.

Reviewer 3

Point 1: It will be important to knock down the Siah2 in the CGNs and examine the effect on the drebrin ubiquitination, protein level, stability, localization and function.

Response: We greatly appreciate the reviewer's request to provide additional supporting data regarding the antagonistic relationship between Siah E3 ligases and Drebrin as we feel the suggested experiments strengthen our manuscript significantly. To address both Points 1 and 4 we provide substantive new data regarding our Drebrin NxN mutant and the affect of Siah mutants on Drebrin protein levels in HEK293 cells and CGNs.

New Figure 7 HEK293 Data: As the reviewer suspected, Drebrin NxN protein levels are less sensitive to both Siah1 and Siah2 overexpression compared to wild type drebrin. Moreover, this mutant shows higher steady state levels in the absence of Siah expression and significantly less ubiquitination in a pull down assay. To address concerns regarding our original Siah2 dRING experiments we turned to additional tools: Siah1 M180K and Siah2 M180K are defined point mutations that inhibit Siah dimerization and substrate binding. Expression of Siah1 M180K and Siah2 M180K leads to significantly less reduction in Drebrin levels than their wild type counterparts.

New Figure 7 CGN data: We found that Siah2-silencing leads to enhanced drebrin expression as predicted by our model. Further, we exploited Drebrin NxN to examine the affects of diminished Siah regulation of drebrin in cultured CGNs. Similar to what we observed in HEK293 cells, Drebrin 2xNxN 2xVenus displays higher steady state protein levels when nucleofected into cultured CGNs. Interestingly, Drebrin 2xNxN 2xVenus also shows a broader leading process distribution and a two-fold increase in FRAP t1/2 compared to wild type Drebrin 2xVenus, suggesting that Siah-mediated drebrin turnover is altered by removal of the degron sequences.

Supplementary Figure 5: To address a functional relationship between Siah2 and drebrin, we performed co-shRNA silencing experiments *ex vivo* to supplement our original Siah2-drebrin gain-of-function experiments. Previous work from our laboratory showed that Siah2, but not Siah1, silencing stimulates precocious CGN germinal zone exit and radial migration. Interestingly, silencing both Siah2 and drebrin inhibits the precocious CGN germinal zone exit and radial migration, suggesting that drebrin activity is required for the altered motility seen in Siah2-silenced CGNs. These data complement our original experiments showing the drebrin NxN expression rescues Siah2 gain-of-function migration phenotypes. Taken together, these results show that the Siah-drebrin antagonistic relationship clearly affects drebrin protein levels, stability, localization and function in the CGN.

Point 2: Is the level, activity or localization of endogenous Siah2 altered during the CGN migration, and under what physiological condition that endogenous Siah2 degrades drebrin and regulates the CNG migration?

Response: Figure 7b shows an accurate representation of Siah2 protein expression at postnatal day seven (P7). Siah2 expression is high in granule neuron progenitors (GNPs), is extinguished in postmitotic CGNs and is not observed in CGNs in the molecular layer or IGL. Additional published work from our laboratory in Famulski et al 2010 show this expression profile using additional CGN differentiation makers like Tuj1 and Tag1 and also show that CGN Siah2 expression is extinguished when the EGL disappears by P15. One physiological explanation for the Siah-drebrin antagonism could be related to differentiation

status. The absence of drebrin expression in GNPs could be the result of high Siah2 expression in GNPs.

To experimentally address the reviewer's question we treated cultured CGNs with sonic hedgehog (Shh), a mitogen that blocks differentiation and GNP GZ exit. Interestingly, our laboratory recently found that Siah2 expression levels are elevated in GNPs maintained by sonic hedgehog treatment (Ong and Solecki, unpublished results). While our Shh-Siah2 project is a separate ongoing study, we provide data in Figure 7 showing that Shh treated GNP-CGN mixed cultures not only leads to elevated levels of Siah2 positive GNPs but also apparently less drebrin immunoreactivity. These results provide correlative support to the hypothesis that differentiation status and competency to radially migrate may be related to the Siah2-drebrin antagonistic relationship.

Point 3: Siah2 knockout mice have no apparent phenotype in the neuron system. It will be interesting to see whether Siah1 expresses in CGNs, and whether both Siah1 and Siah2 are required for the regulation of drebrin and CNG migration.

Published work from our laboratory in Famulski et al 2010 shows a small amount of Siah1 expression in the inner EGL of the P7 cerebellum, which like Siah2, disappears from CGNs when the EGL GZ regresses at P15. Moreover, we showed that Siah1 silencing had little effect on CGN GZ exit and migration. To address the reviewer's comments we carried out similar HEK293 biochemistry and CGN ex vivo epistasis experiments as we described for Siah2 above. We found that wild type Siah1, but not Siah1 M180K, overexpression leads to reduced amounts of drebrin and drebrin NxN 2xVenus is less sensitive to Siah1. In our CGN epistasis experiments, we confirmed that Siah1 silencing does not induce GZ exit and radial migration thus we could not assess a GZ exit epistasis between Siah1 and drebrin. We note that drebrin silencing does not fully inhibit CGN migration at 48 hours of ex vivo culture in the context of Siah1 loss-of-function as we observed with drebrin silencing alone (see Supplementary Figure 5), potentially suggesting a minor functional interaction.

Point 4. The biochemical analysis of Siah2 overexpression on drebrin degradation is insufficient. The western blot in Figure 7c shows that the catalytic-inactive Siah2 ring mutant seems to also reduce the level of GFP-tagged drebrin. The authors need to provide more convincing evidence that Siah2 overexpression indeed changes the ubiquitination and stability of drebrin, and the drebrin NxN mutant shows a reduced interaction with Siah2.

Response: See our comprehensive response to Point 1 with address these concerns.

Point 5. For the functional analysis of Siah2 overexpression or Siah2 overexpression plus Drebrin NxN mutant in Figure7, it will be important to show the western blot using the drebrin antibody, to prove whether Siah2

overexpression decreases the level of endogenous drebrin and whether the expression of the drebrin NxN mutant restores the loss of endogenous drebrin.

Response: We examined whether Siah2 overexpression targets endogenous drebrin for degradation using HEK293 cells. HEK293 cells express high levels of a full-length drebrin and a faster migrating isoform. The highest molecular weight drebrin isoform is significantly diminished by Siah2 expression but the lower molecular weight form is not. This result matches nicely with known drebrin splice variants that lack the c-terminal sequences that harbor the Siah degrons. In this situation, Drebrin 2NxN expression only had a mild affect on the level of endogenous drebrin when Siah2 was expressed. Drebrin 2xNxN lacks Siah degrons and it appears that cannot act as a competitive Siah inhibitor by providing additional degrons for Siah ligases to bind to.

We include an image for the reviewer:

Legend: HEK293 cells were transfected LacZ (lane 1), Siah2 or Siah2 and drebrin 2xNxN 2xVenus expression constructs via LF2000. 24 h post transfection cell lysates were prepared and SDS-PAGE performed. Western blots were probed with mouse rabbit anti-drebrin A/E and beta actin antibodies.

Reviewers' Comments:

Reviewer #1 (Remarks to the Author)

This referee is satisfied about the author's response to the referee's previous comment.

Reviewer #2 (Remarks to the Author)

The authors perform a study intending to investigate microtubule (MT)-actomyosin coupling in the migration of CGNs. They used sophisticated imaging technology to demonstrate MT-actomyosin coupling and they propose drebrin as molecular mediator for this interaction. Some questions were raised in the first round of reviewing.

The authors convincingly answered our questions and they also added new information concerning the mechanisms responsible for drebrin translocation in the cell and further discuss in the manuscript how the interaction of drebrin with MTs takes place.

Overall, the authors greatly improved the quality of the manuscript and clarified our main concerns regarding the interpretation of some experiments.

Reviewer #3 (Remarks to the Author)

The authors have adequately addressed the concerns raised in my previous critiques. It is ready for publication.